# Gridded Transformer Neural Processes for Large Unstructured Spatio-Temporal Data

## Abstract

Many important problems require modelling large-scale spatio-temporal datasets, with one prevalent example being weather forecasting. Recently, transformer-based approaches have shown great promise in a range of weather forecasting problems. However, these have mostly focused on gridded data sources, neglecting the wealth of unstructured, off-the-grid data from observational measurements such as those at weather stations. A promising family of models suitable for such tasks are neural processes (NPs), notably the family of transformer neural processes (TNPs). Although TNPs have shown promise on small spatio-temporal datasets, they are unable to scale to the quantities of data used by state-of-the-art weather and climate models. This limitation stems from their lack of efficient attention mechanisms. We address this shortcoming through the introduction of gridded pseudo-token TNPs which employ specialised encoders and decoders to handle unstructured observations and utilise a processor containing gridded pseudo-tokens that leverage efficient attention mechanisms. Our method consistently outperforms a range of strong baselines on various synthetic and real-world regression tasks involving large-scale data, while maintaining competitive computational efficiency. The real-life experiments are performed on weather data, demonstrating the potential of our approach to bring performance and computational benefits when applied at scale in a weather modelling pipeline.

## 1 Introduction

Many spatio-temporal modelling problems are being transformed by the proliferation of data from *in situ* sensors, remote observations, and scientific computing models. The opportunities presented have led to a surge of interest from the machine learning community to develop new tools and models to support these efforts. One prominent example of this transformation is in medium-range weather and environmental forecasting where a new generation of machine learning models have improved performance and reduced computational costs, including: Aurora (Bodnar et al., 2024); GraphCast (Lam et al., 2022); GenCast (Price et al., 2023); PanguWeather (Bi et al., 2022); ClimaX (Nguyen et al., 2023), FuXi (Chen et al., 2023b); and FengWu (Chen et al., 2023a). All of these models operate on sets of environmental variables which are regularly structured in space and time, allowing them to leverage architectures developed in the vision and language community such as the Vision Transformer (ViT; Dosovitskiy et al. 2020), Swin Transformer (Liu et al., 2021), and Perceiver (Jaegle et al., 2021). These models are trained and deployed on data arising from computationally intensive scientific simulation and analysis techniques, which integrate observational data (e.g. from weather stations, ships, buoys, radiosondes, etc.) with simulation data to provide the best estimate of the atmosphere's state. They are not currently trained on observational measurements directly.

We are now on the cusp of a second generation of these models which will also ingest unstructured observational, alongside analysis data, or which will replace the need for analysis data entirely. This second generation of forecasting systems will further improve accuracy and reduce computational costs. Thus far, there have only been a handful of early attempts to tackle this problem (Vaughan et al., 2024; McNally et al., 2024; Xu et al., 2024; Xiao et al., 2023) and it is unclear what architectures would best support this setting. We therefore consider it fertile ground for impactful research.

Many problems considered can be formulated as repeatedly performing predictive inference conditioned on an ever-changing, large set of observations. Posed with the question of what framework

is suitable for handling spatio-temporal prediction problems containing both structured and unstructured data, we turn towards neural processes (NPs; Garnelo et al. 2018a;b): a family of meta-learning models that are able to map from datasets of arbitrary size and structure to predictions over outputs at arbitrary input locations. NPs support a probabilistic treatment of observations, making them capable of outputting uncertainty estimates that are crucial in, for example, forecasting systems. Moreover, NPs are a flexible framework—unlike other models that only perform forecasting, they are able to solve more general state estimation problems, including forecasting, data fusion, data interpolation and data assimilation. Although early versions of NPs were severely limited, a string of recent developments (Kim et al., 2019; Gordon et al., 2019; Nguyen & Grover, 2022; Ashman et al., 2024a; Feng et al., 2023; Bruinsma et al., 2021; Ashman et al., 2024b) have significantly improved their effectiveness, particularly for small-scale spatio-temporal regression problems. Their broad applicability has been demonstrated throughout literature, from a wide variety of spatio-temporal tasks such as climate downscaling, data assimilation and sensor placement (Vaughan et al., 2022; Andersson et al., 2023; Chen et al., 2024; Niu et al., 2024), to tasks as diverse as molecular property prediction (García-Ortegón et al., 2024). Notably, the family of transformer NPs (TNPs; Nguyen & Grover 2022; Feng et al. 2023; Ashman et al. 2024a), which use transformer-based architectures as the computational backbone, have demonstrated impressive performance on a range of tasks. However, unlike the aforementioned large-scale environmental prediction models, TNPs are yet to fully take advantage of efficient attention mechanisms. As a result of the quadratic computational complexity associated with full attention, they have been unable to scale to complex spatio-temporal datasets. This is because such techniques require structured—more specifically, gridded—datasets, and are thus not immediately applicable.

We pursue a straightforward solution: to encode the dataset onto a structured grid before passing it through a transformer-based architecture. We refer to this family of models as *gridded TNPs*, serving as a general-purpose tool for spatio-temporal state estimation. Our core contributions are:

1. We develop a novel attention-based grid encoder, based on the ideas of 'pseudo-tokens' (Jaegle et al., 2021; Feng et al., 2023; Lee et al., 2019), which we show improves upon the performance of traditional kernel-based interpolation methods.

2. Equipped with the pseudo-token grid encoder, we are able to use efficient attention mechanisms within the transformer backbone of the TNP, utilising advancements such as the ViT (Dosovitskiy et al., 2020) and Swin Transformer (Liu et al., 2021).

3. We develop an efficient $k$-nearest-neighbour attention-based grid decoder, facilitating the evaluation of predictive distributions at arbitrary spatio-temporal locations. Remarkably, we find that this improves performance over full attention.

4. We empirically evaluate our model on a range of synthetic and real-world spatio-temporal regression tasks, demonstrating both the ability to 1) maintain strong performance on large spatio-temporal datasets and 2) handle multiple sources of unstructured data effectively, all while maintaining a low computational complexity.

## 2 BACKGROUND

We consider the supervised learning setting, where $\mathcal{X}$, $\mathcal{Y}$ denote the input and output spaces, and $(\mathbf{x}, \mathbf{y}) \in \mathcal{X} \times \mathcal{Y}$ denotes an input-output pair. We restrict our attention to $\mathcal{X} = \mathbb{R}^{D_x}$ and $\mathcal{Y} = \mathbb{R}^{D_y}$. Let $\mathcal{S} = \bigcup_{N=0}^{\infty} (\mathcal{X} \times \mathcal{Y})^N$ be a collection of all finite datasets, which includes the empty set $\varnothing$. We denote a context and target set with $\mathcal{D}_c$, $\mathcal{D}_t \in \mathcal{S}$, where $|\mathcal{D}_c| = N_c$, $|\mathcal{D}_t| = N_t$. Let $\mathbf{X}_c \in (\mathcal{X})^{N_c}$, $\mathbf{Y}_c \in (\mathcal{Y})^{N_c}$ be the inputs and corresponding outputs of $\mathcal{D}_c$, and let $\mathbf{X}_t \in (\mathcal{X})^{N_t}$, $\mathbf{Y}_t \in (\mathcal{Y})^{N_t}$ be defined analogously. We denote a single task as $\xi = (\mathcal{D}_c, \mathcal{D}_t) = ((\mathbf{X}_c, \mathbf{Y}_c), (\mathbf{X}_t, \mathbf{Y}_t))$.

### 2.1 NEURAL PROCESSES

Neural processes (NPs; Garnelo et al. 2018a;b) can be viewed as neural-network-based mappings from context sets $\mathcal{D}_c$ to predictive distributions at target locations $\mathbf{X}_t$, $p(\cdot \mid \mathbf{X}_t, \mathcal{D}_c)$. In this work, we restrict our attention to conditional NPs (CNPs; Garnelo et al. 2018a), which only target marginal predictive distributions by assuming that the predictive densities factorise: $p(\mathbf{Y}_t|\mathbf{X}_t, \mathcal{D}_c) = \prod_{n=1}^{N_t} p(\mathbf{y}_{t,n}|\mathbf{x}_{t,n}, \mathcal{D}_c)$. We denote all parameters of a CNP by $\theta$. CNPs are

trained in a meta-learning fashion, in which the expected predictive log-probability is maximised:

$$\theta_{\mathrm{ML}} = \arg\max_\theta \mathcal{L}_{\mathrm{ML}}(\theta) \quad \text{where} \quad \mathcal{L}_{\mathrm{ML}}(\theta) = \mathbb{E}_{p(\xi)}\big[\textstyle\sum_{n=1}^{N_t} \log p_\theta(\mathbf{y}_{t,n}|\mathbf{x}_{t,n}, \mathcal{D}_c)\big]. \tag{1}$$

For real-world datasets, we only have access to a finite number of tasks for training and so approximate this expectation with an average over tasks. The global maximum is achieved if and only if the model recovers the ground-truth predictive distributions (Proposition 3.26 by Bruinsma, 2022). In Appendix A we present a unifying construction for CNPs involving three components: the *encoder* $e\colon \mathcal{X} \times \mathcal{Y} \to \mathcal{Z}$, which encodes each $(\mathbf{x}_{c,n}, \mathbf{y}_{c,n}) \in \mathcal{D}_c$ into some token representation $\mathbf{z}_{c,n} \in \mathcal{Z}$, the *processor* $\rho\colon \left(\bigcup_{n=0}^\infty \mathcal{Z}^n\right) \times \mathcal{X} \to \mathcal{Z}$, which processes the set of context tokens together with the target input $\mathbf{x}_t$ to obtain a target dependent token $\mathbf{z}_t \in \mathcal{Z}$, and the *decoder* $d\colon \mathcal{Z} \to \mathcal{P}_\mathcal{Y}$, which maps from the target token to the predictive distribution over the output at that target location. Here, $\mathcal{P}_\mathcal{Y}$ denotes the space of distributions over the output space $\mathcal{Y}$.

## 2.2 Transformers and Transformer Neural Processes

A useful perspective of transformers is that of set functions (Lee et al., 2019), making them suitable in the construction of the processor of a CNP, resulting in the family of TNPs. In this section, we provide a brief overview of transformers and their use in TNPs.

### 2.2.1 Self-Attention and Cross-Attention

Broadly speaking, transformer-based architectures consist of two operations: multi-head self-attention (MHSA) and multi-head cross-attention (MHCA). Informally, the MHSA operation can be understood as updating a set of tokens using the same set, whereas the MHCA operation can be understood as updating one set of tokens using a different set. More formally, let $\mathbf{Z} \in \mathbb{R}^{N \times D_z}$ denote a set of $N$ $D_z$-dimensional input tokens. The MHSA operation updates this set of tokens as

$$\mathbf{z}_n \leftarrow \mathrm{cat}\left(\left\{\textstyle\sum_{m=1}^N \alpha_h(\mathbf{z}_n, \mathbf{z}_m)\mathbf{z}_m{}^T \mathbf{W}_{V,h}\right\}_{h=1}^H\right)\mathbf{W}_O \quad \forall\, n = 1, \dots, N. \tag{2}$$

Here, $\mathbf{W}_{V,h} \in \mathbb{R}^{D_z \times D_V}$ and $\mathbf{W}_O \in \mathbb{R}^{HD_V \times D_z}$ are the value and projection weight matrices, where $H$ denotes the number of 'heads', and $\alpha_h$ is the attention mechanism. This is most often a softmax-normalised transformed inner-product between pairs of tokens: $\alpha_h(\mathbf{z}_n, \mathbf{z}_m) = \mathrm{softmax}(\{\mathbf{z}_n^T \mathbf{W}_{Q,h} \mathbf{W}_{K,h}^T \mathbf{z}_m\}_{m=1}^N)_m$, where $\mathbf{W}_{Q,h} \in \mathbb{R}^{D_z \times D_{QK}}$ and $\mathbf{W}_{K,h} \in \mathbb{R}^{D_z \times D_{QK}}$ are the query and key matrices. The MHCA operation updates one set of tokens, $\mathbf{Z}_1 \in \mathbb{R}^{N_1 \times D_z}$, using another set of tokens, $\mathbf{Z}_2 \in \mathbb{R}^{N_2 \times D_z}$, in a similar manner:

$$\mathbf{z}_{1,n} \leftarrow \mathrm{cat}\left(\left\{\textstyle\sum_{m=1}^{N_2} \alpha_h(\mathbf{z}_{1,n}, \mathbf{z}_{2,m})\mathbf{z}_{2,m}{}^T \mathbf{W}_{V,h}\right\}_{h=1}^H\right)\mathbf{W}_O \quad \forall\, n = 1, \dots, N_1. \tag{3}$$

MHSA and MHCA operations are used in combination with layer-normalisation operations and point-wise MLPs to obtain MHSA and MHCA blocks. Unless stated otherwise, we shall adopt the order used by Vaswani et al. (2017) which we detail in Appendix E.

### 2.2.2 Pseudo-Token-Based Transformers

Pseudo-token-based transformers, first introduced by Jaegle et al. (2021) with the Perceiver, remedy the quadratic computational complexity induced by the standard transformer by condensing the set of $N$ tokens, $\mathbf{Z} \in \mathbb{R}^{N \times D_z}$, into a smaller set of $M \ll N$ 'pseudo-tokens', $\mathbf{U} \in \mathbb{R}^{M \times D_z}$, using MHCA operations.[1] These pseudo-tokens are then processed instead of operating on the original set directly, reducing the computational complexity from $\mathcal{O}(N^2)$—the cost of performing MHSA operations on the original set—to $\mathcal{O}(NM + M^2)$—the cost of performing MHCA operations between the original set and pseudo-tokens, followed by MHSA operations on the pseudo-tokens.

### 2.2.3 Transformer Neural Processes

Transformer neural processes (TNPs) use transformer-based architectures as the processor in the CNP construction described in Section 2.1. First, each context point $(\mathbf{x}_{c,n}, \mathbf{y}_{c,n}) \in \mathcal{D}_c$ and target

---

[1]There are strong similarities between the use of pseudo-tokens in transformers and the use of inducing points in sparse Gaussian processes: both can be interpreted as condensing the original dataset into a smaller, 'summary dataset'.

input $\mathbf{x}_{t,n} \in \mathbf{X}_t$ is encoded to obtain an initial set of context and target tokens, $\mathbf{Z}_c^0 \in \mathbb{R}^{N_c \times D_z}$ and $\mathbf{Z}_t^0 \in \mathbb{R}^{N_t \times D_z}$. Transformers are then used to process the union $\mathbf{Z}^0 = \mathbf{Z}_c^0 \cup \mathbf{Z}_t^0$ using a series of MHSA and MHCA operations, keeping only the output tokens corresponding to the target inputs. These processed target tokens are then mapped to predictive distributions using the decoder. The specific transformer-based architecture is unique to each TNP variant, with each generally consisting of MHSA operations acting on the context tokens—or pseudo-token representation of the context— and MHCA operations acting to update the target tokens given the context tokens. We provide an illustrative diagram of two popular TNP variants in Appendix A: the regular TNP (Nguyen & Grover, 2022) and the induced set transformer NP (ISTNP; Lee et al. 2019).

The application of TNPs to large datasets is impeded by the use of MHSA and MHCA operations acting on the entire set of context and target tokens. Even when pseudo-tokens are used in pseudo-token-based TNPs (PT-TNPs), the number of pseudo-tokens $M$ required for accurate predictive inference generally scales with the complexity of the dataset, and the MHCA operations between the pseudo-tokens and the context and target tokens quickly becomes prohibitive as the size of the dataset increases. This motivates the use of efficient attention mechanisms, as used in the ViT (Dosovitskiy et al., 2020) and Swin Transformer (Liu et al., 2021).

## 3 RELATED WORK

**Transformers for Point Cloud Data** A closely related research area is point cloud data modelling (Tychola et al., 2024), with transformer-based architectures employed for a variety of tasks (Lu et al., 2022). Given the large amount of data used in some works, the use of efficient attention mechanisms has been considered. A number of notable approaches use voxelisation, whereby unstructured point clouds are encoded onto a structured grid (Mao et al., 2021; Zhang et al., 2022), prior to employing efficient architectures that operate on grids. Both aforementioned works voxelise the unstructured inputs through rasterisation, but implement efficient self-attention mechanisms such as Swin Transformer. Our pseudo-token grid encoder is heavily inspired by the voxel-based set attention (VSA) introduced by He et al. (2022). Similar to our approach, VSA cross-attends local neighbourhoods of unstructured tokens onto a structured grid of pseudo-tokens. However, VSA uses the same initial pseudo-token values for all grid locations, as the 'cloud' in which points exist are not equipped with unobserved information. Their use of a fixed set of initial values manifests a specific implementation in which all tokens attend to the same set of pseudo-tokens. In contrast, for many spatio-temporal problems there may exist potentially unobserved, fixed topographical information such as elevation, land use and soil type, hence we employ different initial pseudo-token values for each grid location which are capable of capturing this.

**Models for Structured Weather Data** The motivation behind our method is to develop models that are able to scale to massive spatio-temporal datasets. In recent years, doing so has garnered a significant amount of interest from the research community, and several models have been developed—some of the most successful of which target weather data. However, the majority of these methods operate on structured, gridded data, and focus solely on forecasting.[2] Models such as Aurora (Bodnar et al., 2024), GraphCast (Lam et al., 2022), GenCast (Price et al., 2023), Pangu (Bi et al., 2022), FuXi (Chen et al., 2023b) and FengWu (Chen et al., 2023a) share a similar encoder-processor-decoder to our construction of CNPs presented in Section 2.1, in which the input grid is projected onto some latent space which is then transformed using an efficient form of information propagation (e.g. sparse attention with Swin Transformer (Bodnar et al., 2024; Bi et al., 2022), message passing with GNNs (Price et al., 2023)) before being projected back onto the grid at the output.[3]

Although yielding impressive performance, these methods are inherently constrained in their application—forecasting with structured data—with predictions sharing the same locations as the inputs at each time point. The distinct advantage of NPs is their ability to model stochastic processes, which can be evaluated at any target location, and their ability to flexibly condition on un-

---

[2]That is, to predict the entire gridded state at time $t + 1$ given a history of previous gridded state(s).

[3]It is worth noting that despite transformers being associated with fully-connected graphs, sparse attention mechanisms such as Swin Transformer can also be interpreted as a form of sparsely connected GNN with staggered updates.

structured data. Together, these enable NPs to tackle a strictly larger class of tasks. Indeed, our work reflects a more general trend towards more flexible methods for spatio-temporal data, particularly those that are able to deal with unstructured data. Perhaps the most relevant works to ours are Aardvark (Vaughan et al., 2024) and FuXi-DA (Xu et al., 2024), both end-to-end weather prediction models that handle both unstructured and structured data. Aardvark employs kernel interpolation, also known as a SetConv (Gordon et al., 2019), to move unstructured data onto a grid, followed by a ViT which processes this grid and outputs gridded predictions, whereas FuXi-DA simply averages observations within each grid cell. We provide an extensive comparison between the kernel interpolation approach to structuring data and our pseudo-token grid encoder in Section 5, demonstrating significantly better performance. Another relevant example is Lessig et al. (2023), a task-agnostic stochastic model of atmospheric dynamics, trained to predict randomly masked or distorted tokens. While the task agnosticism of this approach shares similarities with that of NPs, it is unclear how this approach extends to settings in which data can exist at arbitrary spatio-temporal locations, and they are more limited in their approach to model multiple sources of the input data.

## 4 GRIDDED TRANSFORMER NEURAL PROCESSES

While TNPs have demonstrated promising performance on small to medium-sized datasets, they are unable to scale to the size characteristic of large spatio-temporal datasets. To address this shortcoming, we consider the use of efficient attention mechanisms that have proved effective for gridded spatio-temporal data (Bodnar et al., 2024; Lam et al., 2022; Price et al., 2023; Nguyen et al., 2023; Bi et al., 2022). Such methods are not immediately applicable without first structuring the unstructured data. We achieve this by drawing upon methods developed in the point cloud modelling literature— notably the voxel-based set attention (VSA) (He et al., 2022)—and develop the pseudo-token grid encoder: an effective attention-based method for encoding unstructured data onto a grid.

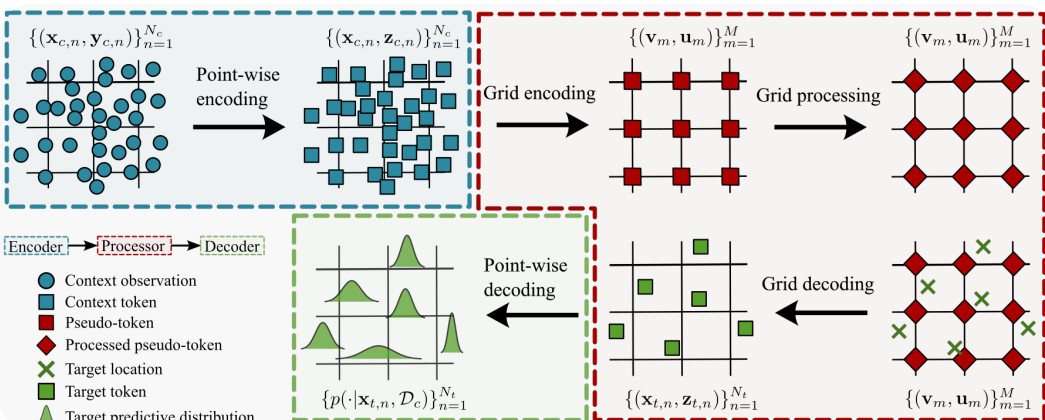

**Figure 1:** An illustrative demonstration of the complete gridded TNP pipeline. Following the CNP constrution in Section 2.1, we highlight the encoder (blue), processor (red) and decoder (green).

We provide an illustrative diagram of our proposed approach in Figure 1, which decomposes the processor $\rho \colon \left( \bigcup_{n=0}^{\infty} \mathcal{Z}^n \right) \times \mathcal{X} \to \mathcal{Z}$ into three parts: 1. the grid encoder, $\rho_{ge} \colon \bigcup_{n=0}^{\infty} \mathcal{Z}^n \to \mathcal{Z}^M$, which embeds the set $\left\{ (\mathbf{x}_{c,n}, \mathbf{z}_{c,n} \right\}_{n=1}^{N_c}$ into tokens $\{\mathbf{u}_m\}_{m=1}^{M}$ at gridded locations $\{\mathbf{v}_m\}_{m=1}^{M}$; 2. the grid processor, $\rho_{gp} \colon \mathcal{Z}^M \to \mathcal{Z}^M$, which transforms the token values; and 3. the grid decoder, $\rho_{gd} \colon \mathcal{Z}^M \times \mathcal{X} \to \mathcal{Z}$, which maps the tokens onto the location $\mathbf{x}_t$. Here, the latent space $\mathcal{Z} = \mathcal{Z}_{\text{token}} \times \mathcal{X}$. We discuss our choices for each of these components in the remainder of this section.

### 4.1 GRID ENCODER: THE PSEUDO-TOKEN GRID ENCODER

Let $\mathbf{z}_n \in \mathbb{R}^{D_z}$ denote the token representation of input-output pair $(\mathbf{x}_n, \mathbf{y}_n)$ after point-wise embedding. We introduce a set of $\prod_{d=1}^{D_x} M_d$ pseudo-tokens $\mathbf{U}^0 \in \mathbb{R}^{M_1 \times \cdots \times M_{D_x} \times D_z}$ at corresponding locations on the grid $\mathbf{V} \in \mathbb{R}^{M_1 \times \cdots M_{D_x} \times D_x}$. That is, pseudo-token $\mathbf{u}_{m_1, \ldots, m_{D_x}} \in \mathbb{R}^{D_z}$ is associated with location $\mathbf{v}_{m_1, \ldots, m_{D_x}} \in \mathbb{R}^{D_x}$. For ease of reading, we shall replace the product $\prod_{d=1}^{D_x} M_d$

with $M$ and the indexing notation $m_1, \ldots, m_{D_x}$ with $m$. The pseudo-token grid encoder obtains a pseudo-token representation $\mathbf{U} \in \mathbb{R}^{M \times D_z}$ of $\mathcal{D}_c$ on the grid $\mathbf{V}$ by cross-attending from each set of tokens $\{\mathbf{z}_{c,n}\}_{n \in \mathfrak{N}(\mathbf{v}_m; k)}$ to each initial pseudo-token $\mathbf{u}_m^0$:

$$\mathbf{u}_m \leftarrow \mathrm{MHCA}\left(\mathbf{u}_m^0, \{\mathbf{z}_{c,n}\}_{n \in \mathfrak{N}(\mathbf{v}_m; k)}\right) \quad \forall m \in M. \tag{4}$$

Here, $\mathfrak{N}(\mathbf{v}_m; k)$ denotes the index set of input locations for which $\mathbf{v}_m$ is amongst the $k$ nearest grid locations. In practice, we found that $k = 1$ suffices. While this operation seems computationally intensive, there are two tricks we can apply to make it computationally efficient. First, note that provided the pseudo-token grid is regularly spaced, the set of nearest neighbours for all grid locations can be found in $\mathcal{O}(N_c)$. Second, the operation described in Equation 4 can be performed in parallel for all $m \in M$ by 'padding' each set $\{\mathbf{z}_{c,n}\}_{n \in \mathfrak{N}(\mathbf{v}_m; k)}$ with $\max_i |\mathfrak{N}(\mathbf{v}_i; k)| - |\mathfrak{N}(\mathbf{v}_m; k)|$ 'dummy tokens' and applying appropriate masking, resulting in a computational complexity of $\mathcal{O}\left(M \max_i |\mathfrak{N}(\mathbf{v}_i; k)|\right)$. Restrictions can be placed on the size of each neighbourhood set to reduce this further. We illustrate the pseudo-token grid encoding in Figure 2.

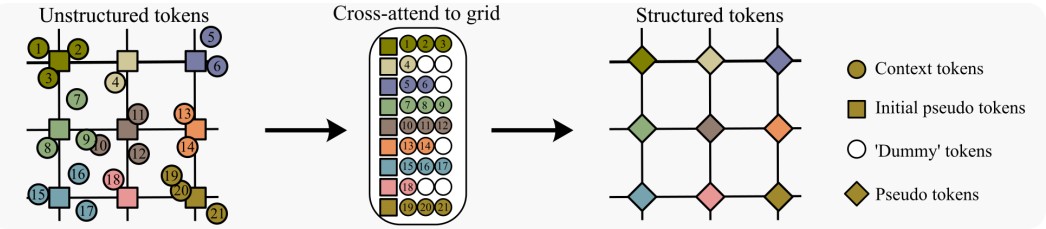

**Figure 2:** An illustrative demonstration of the pseudo-token grid encoder in the 2-D case. To achieve an efficient implementation of cross-attention to the pseudo-token grid, we pad sets of neighbourhood tokens with 'dummy' tokens, so that each neighbourhood has the same cardinality.

In our experiments, we compare the performance of the pseudo-token grid encoder with simple kernel-interpolation onto a grid, a popular method used by the ConvCNP. We provide a detailed description of this approach in Appendix B.

### 4.2 Grid Processor: Efficient Attention Mechanism-Based Transformers

While we are free to choose any architecture for processing the pseudo-token grid, including CNNs, we focus on transformer-based architectures employing efficient attention mechanisms. Specifically, we consider the use of ViT (Dosovitskiy et al., 2020) and Swin Transformer (Liu et al., 2021). Typically, ViT employs *patch encoding* to coarsen a grid of tokens to a coarser grid of tokens, upon which regular MHSA operations are applied. We consider both patch encoding and encoding directly to a smaller grid using the grid encoder, which we collectively refer to when used in gridded TNPs as ViTNPs. The use of Swin Transformer has the advantage of being able to operate with a finer grid of pseudo-tokens, as only local neighbourhoods of tokens attend to each other at any one time. We refer to the use of Swin Transformer in gridded TNPs as Swin-TNP.

### 4.3 Grid Decoder: the Cross-Attention Grid Decoder

In the PT-TNP, *all* pseudo-tokens $\mathbf{U}$ cross-attend to *all* target tokens $\mathbf{Z}_t$. This has a computational complexity of $\mathcal{O}(MN_t)$ which is prohibitive for large $N_t$ and $M$. We propose nearest-neighbour cross-attention, in which the set of pseudo-tokens $\{\mathbf{u}_m\}_{m \in \tilde{\mathfrak{N}}(\mathbf{x}_{t,n}; k)}$ attend to the target token $\mathbf{z}_{t,n}$:

$$\mathbf{z}_{t,n} \leftarrow \mathrm{MHCA}\left(\mathbf{z}_{t,n}^0, \{\mathbf{u}_m\}_{m \in \tilde{\mathfrak{N}}(\mathbf{x}_{t,n}; k)}\right). \tag{5}$$

Here, $\tilde{\mathfrak{N}}(\mathbf{x}_{t,n}; k)$ denotes the index set of the $k$ grid locations that are closest to $\mathbf{x}_{t,n}$.[4] Assuming equally spaced grid locations, this set can be computed for all target tokens in $\mathcal{O}(N_t)$, and the operation in Equation 5 can be performed in parallel with a computational complexity of $\mathcal{O}(kN_t)$.

---

[4]More specifically, if the dimension of the target inputs is $d = \dim(x_{t,n})$ for $\forall t, n$, we choose the neighbours within the hypercube defined by the nearest $k^{1/d}$ neighbours in each dimension. When $k^{1/d}$ is not an integer, we take the smallest integer bigger than it (i.e. $\mathrm{ceil}(k^{1/d})$). For more details, please refer to Appendix C.

For $k \ll M$, this represents a significant improvement in computational complexity. Further, we found that the use of nearest-neighbour cross-attention improved upon full cross-attention in practice, which we attribute to the inductive biases it introduces being useful for spatio-temporal data.

### 4.4 HANDLING MULTIPLE DATA MODALITIES

Spatio-temporal datasets often have multiple sources of data, and multi-modal data fusion is an active area of research. It is possible to extend our method to such scenarios when the sources share a common input domain $\mathcal{X}$, the most straightforward of which is to use a different encoder for each modality and apply the pseudo-token grid encoder to the union of tokens. Formally, let $\mathcal{D}_c = \bigcup_{s=1}^{S} \mathcal{D}_{c,s}$ denote the context dataset partitioned into $S$ smaller datasets, one for each source. For each source, we obtain $\mathbf{z}_{c,n,s} = e_s(\mathbf{x}_{c,n,s}, \mathbf{y}_{c,n,s}) \ \forall \ (\mathbf{x}_{c,n,s}, \mathbf{y}_{c,n,s}) \in \mathcal{D}_\mathbf{s}, \ \mathbf{s} \in \mathbf{S}$, where $e_s \colon \mathcal{X} \times \mathcal{Y}_s \to \mathcal{Z}$ denotes the source-specific point-wise encoder and $\mathcal{Y}_s$ denotes the output space for source $s$. We obtain the set of context tokens as $\mathbf{Z}_c = \{\{\mathbf{z}_{c,n,s}\}_{n=1}^{N_{c,s}}\}_{s=1}^{S}$. We refer to this approach as the *single* pseudo-token grid encoder. We also consider a second, more involved approach in which a pseudo-token grid is formed independently for each data modality, with these pseudo-token grids then being merged into a single pseudo-token grid. That is, for each set of source-specific context tokens $\mathbf{Z}_{c,s} = \{e_s(\mathbf{x}_{c,n,s}, \mathbf{y}_{c,n,s})\}_{n=1}^{N_{c,s}}$, we obtain a gridded pseudo-token representation $\mathbf{U}_s \in \mathbb{R}^{M \times D_z}$. A unified gridded pseudo-token representation is obtained by passing their concatenation point-wise through some function. We refer to this approach as the *multi* pseudo-token grid encoder. Note that both methods can be applied analogously to the kernel-interpolation grid encoder.

## 5 EXPERIMENTS

In this section, we evaluate the performance of our gridded TNPs on synthetic and real-world regression tasks involving datasets with large numbers of datapoints. We demonstrate that gridded TNPs consistently outperform baseline methods, particularly when the difficulty—in terms of both dataset size and complexity—of the predictive inference task increases, while maintaining a low computational complexity. Throughout, we compare the performance of gridded TNPs with two different grid encoders (GEs)—the kernel-interpolation grid encoder (KI-GE) and our pseudo-token grid encoder (PT-GE)—and two different grid processors—Swin Transformer and ViT. We also make comparisons to the following baselines: the CNP (Garnelo et al., 2018a), the PT-TNP using the induced set-transformer architecture (Lee et al., 2019; Ashman et al., 2024a), and the ConvCNP (Gordon et al., 2019). We do not compare to the regular ANP or TNP (Kim et al., 2019; Nguyen & Grover, 2022) as they are unable to scale to the context set sizes we consider. We provide complete experimental details, including model architectures, datasets and hardware, in Appendices D and E.

### 5.1 META-LEARNING GAUSSIAN PROCESS REGRESSION

We begin with two synthetic 2-D regression tasks with datasets drawn from a Gaussian process (GP) with a squared-exponential (SE) kernel approximated using structure kernel interpolation (SKI) (Wilson & Nickisch, 2015). Each dataset is a sample of $1.1 \times 10^4$ datapoints, with input values sampled uniformly from $\mathcal{U}_{[-6,6]}$. For each dataset, we set $N_c = 1 \times 10^4$ context and $N_t = 1 \times 10^3$ target points. We consider two different setups: one for which the SE kernel lengthscale is fixed to $\ell = 0.1$, implying roughly $1.44 \times 10^4$ 'wiggles' in each sampled dataset, and one for which the lengthscale is fixed to $\ell = 0.5$, implying roughly 576 'wiggles'. While both setups are non-trivial by the standard of typical NP synthetic experiments, we anticipate the former to be intractable for almost all NP variants—it requires the model to be able to both assimilate finely grained information and modulate its predictions accordingly. In Figure 3, we plot the test log-likelihoods against the time taken to complete a single forward pass (FPT) for a number of gridded TNPs using different grid sizes. We compare the performance to four strong baselines: the PT-TNP with $M = 128$ and $M = 256$ pseudo-tokens, and the ConvCNP with the same grid sizes as the Swin-TNP.[5] A table of results is provided in Appendix E.1. Our results demonstrate that gridded TNPs are able to outperform all baselines—particularly for the more complex datasets, for which all gridded TNP models perform significantly better than the strongest ConvCNP baseline—

---

[5]Training and inference for all models is performed on a single NVIDIA GeForce RTX 2080 Ti.

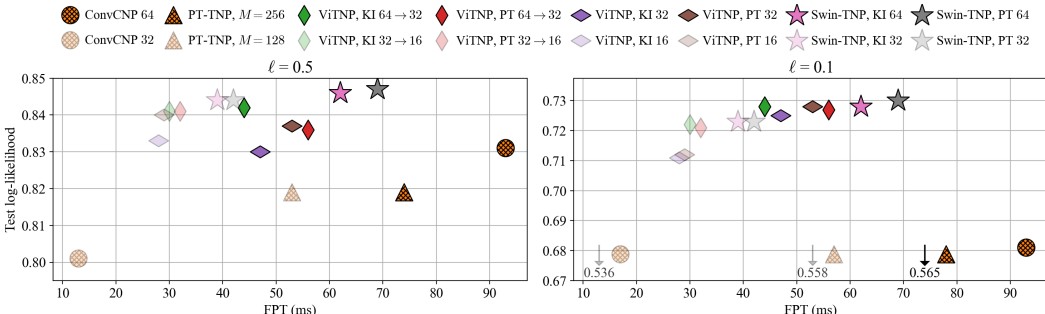

**Figure 3:** Plots comparing the test log-likelihood vs. forward pass time (FPT) for the two synthetic GP datasets. For each model, we show the results for a large and small (transparent) version. The baselines have hatched markers. The grid sizes we consider are $64 \times 64$ and $32 \times 32$, shown as 64 and 32. For the ViTNP models, we include results with and without patch encoding, the former indicated by the $\rightarrow$ symbol in-between the pre- and post-patch-encoded grid sizes. We make use of the following acronyms. KI: kernel-interpolation grid encoding. PT: pseudo-token grid encoding.

while maintaining competitive computational complexity. Further, they suggest that the PT-GE demonstrates superiors performance compared to the KI-GE. In Appendix E.1, we visualise the predictive means of a selection of models for an example dataset with $\ell = 0.1$, which demonstrate the superior ability of the gridded TNPs to capture the complexity of the ground-truth dataset.

## 5.2 COMBINING WEATHER STATION OBSERVATIONS WITH STRUCTURED REANALYSIS

In this experiment, we explore the utility of our model in combining unstructured and sparsely sampled observations with structured, on-the-grid data. We use ERA5 reanalysis data from the European Centre for Medium-Range Weather Forecasts [ECMWF; Hersbach et al. 2020]. We consider two variables: skin temperature (skt) and 2m temperature (t2m).[6] We construct each context dataset by combining the t2m at a random subset of $9,957$ weather station locations (proportion sampled from $\mathcal{U}_{[0,0.3]}$) with a coarsened $180 \times 360$ grid—corresponding to a grid spacing of $1°$—of skt values. The target dataset contains the t2m values at all $9,957$ weather station locations. We train on hourly data between 2009-2017, validate on 2018 and provide test metrics on 2019. Training and inference for all models is performed using a single NVIDIA A100 80GB with 32 CPU cores.

In Table 1, we evaluate the performance of six gridded TNPs. For the Swin-TNP variants, we provide models both with and without the nearest-neighbour cross-attention (NN-CA) mechanism in the decoder. For the Swin-TNP with the PT-GE, we also compare to a model using either solely t2m or skt measurements. Moreover, we provide comparisons to the following baselines: the ConvCNP with a $64 \times 128$ grid using a U-Net (Ronneberger et al., 2015) backbone (and full decoder attention), and the PT-TNP with $M = 256$ pseudo-tokens.[7] Finally, to study how the models scale with model size, we provide results for Swin-TNP (with PT-GE) and ConvCNP with a grid size of $192 \times 384$. For this experiment, we find that the use of patch encoding in the ViTNP leads to better performance than without. We also observe that the Swin-TNP benefits to a greater extent than the ConvCNP when the grid size is increased, and that using the nearest-neighbour cross-attention (NN-CA) mechanism in the grid decoder actually *improves* the performance of the Swin-TNP relative to full cross attention (no NN-CA).

Figure 4 provides a visual comparison between the t2m predictive errors (i.e. difference between predictive mean and ground truth) of three models: the Swin-TNP with the PT-GE, ConvCNP, and PT-TNP. We show the results for the US, and provide results for the world in Figure 18. The most prominent difference between the predictions is at the stations within central US, where both Con-

---

[6]Skin temperature corresponds to the temperature of the Earth's surface, and is a direct output of standard atmospheric models. In contrast, 2m temperature is found by interpolating between the lowest model temperature (10m) and the skin temperature (Owens & Hewson, 2018).

[7]This is the maximum number of pseudo-tokens we could use without running out of memory.

**Table 1:** Test log-likelihood (↑) and RMSE (↓) for the `t2m` station prediction experiment. The standard errors of the log-likelihood are all below $0.010$, and of the RMSE below $0.013$. FPT: forward pass time for a batch size of eight in ms. Params: number of model parameters in units of M.

| Model | GE | Grid size | Log-lik. ↑ | RMSE ↓ | FPT | Params |
|---|---|---|---|---|---|---|
| CNP | - | - | 0.636 | 3.266 | 32 | 0.34 |
| PT-TNP | - | $M = 256$ | 1.344 | 1.659 | 230 | 1.57 |
| ConvCNP (no NN-CA) | SetConv | $64 \times 128$ | 1.535 | 1.252 | 96 | 9.36 |
| ViTNP | KI-GE | $48 \times 96$ | 1.628 | 1.197 | 167 | 1.14 |
| ViTNP | PT-GE | $48 \times 96$ | 1.704 | 1.118 | 181 | 1.83 |
| ViTNP | KI-GE | $144 \times 288 \rightarrow 48 \times 96$ | 1.734 | 1.073 | 171 | 1.29 |
| ViTNP | PT-GE | $144 \times 288 \rightarrow 48 \times 96$ | 1.808 | 1.021 | 215 | 6.69 |
| Swin-TNP | KI-GE | $64 \times 128$ | 1.683 | 1.157 | 121 | 1.14 |
| Swin-TNP | PT-GE | $64 \times 128$ | **1.819** | **1.006** | 127 | 2.29 |
| Swin-TNP (no NN-CA) | KI-GE | $64 \times 128$ | 1.544 | 1.436 | 137 | 1.14 |
| Swin-TNP (no NN-CA) | PT-GE | $64 \times 128$ | 1.636 | 1.273 | 144 | 2.29 |
| Swin-TNP (`skt`) | PT-GE | $64 \times 128$ | 1.427 | 1.330 | 123 | 2.29 |
| Swin-TNP (`t2m`) | PT-GE | $64 \times 128$ | 1.585 | 1.599 | 107 | 2.24 |
| ConvCNP | SetConv | $192 \times 384$ | 1.689 | 1.166 | 74 | 9.36 |
| Swin-TNP | PT-GE | $192 \times 384$ | **2.053** | **0.873** | 306 | 10.67 |

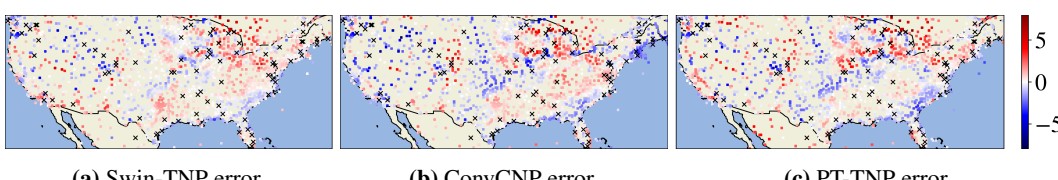

**(a)** Swin-TNP error.   **(b)** ConvCNP error.   **(c)** PT-TNP error.

**Figure 4:** A comparison between the predictive error of the 2m temperature at the US weather station locations at 15:00, 28-01-2019. Stations included in the context dataset are shown as black crosses ($\approx 3\%$ of station locations). The Swin-TNP uses the PT-GE. Both the Swin-TNP and ConvCNP use a grid size of $64 \times 128$. The PT-TNP uses $M = 256$ pseudo-tokens. The mean log-likelihoods of this sample for the three models are $1.611$, $1.351$, and $1.271$, respectively.

vCNP and PT-TNP tend to underestimate the temperature, whereas the Swin-TNP has relatively small predictive errors. We perform an analysis of the predictive uncertainties in Appendix E.2.1.

### 5.3 COMBINING MULTIPLE SOURCES OF UNSTRUCTURED WIND SPEED OBSERVATIONS

In our final experiment, we evaluate the utility of our gridded TNPs in modelling sparsely sampled eastward (`u`) and northward (`v`) components of wind at three different pressure levels: 700hPa, 850hPa and 1000hPa (surface level). Each of the six modalities are obtained from the ERA5 reanalysis dataset (Hersbach et al., 2020). We perform spatio-temporal interpolation over a latitude / longitude range of $[25°, 49°]$ / $[−125°, −66°]$, which corresponds to the contiguous US, spanning four hours. The proportion of the $543,744$ observations used as the context dataset is sampled from $\mathcal{U}_{[0.05,0.25]}$. The target set size is fixed at $135,936$ (i.e. $25\%$ of observations).

We evaluate the performance of eight gridded TNPs: the ViTNP with a $4 \times 12 \times 30$ grid of pseudo-tokens using the multi grid-encoding method presented in Section 4.4, both the KI-GE and PT-GE, and with and without patch encoding from a $4 \times 24 \times 60$ grid; and the Swin-TNP with a $4 \times 24 \times 60$ grid of pseudo-tokens using both multi-modal grid-encoding methods presented in Section 4.4, and both the KI-GE and PT-GE. We provide comparisons to the following baselines: the ConvCNP[8] with a $4 \times 24 \times 60$ grid using a U-Net backbone, and the PT-TNP with $M = 64$ pseudo-tokens.[9] We also provide results for the ConvCNP and Swin-TNP with the PT-GE with larger grid sizes. We visualise the predictive errors for the Swin-TNP, ConvCNP and PT-TNP in Appendix E.3, which

---

[8]For the ConvCNP, each modality is treated as a different input channel as in Vaughan et al. (2024).

[9]We found this to be the maximum number of pseudo-tokens we could use before running out of memory.

demonstrate the Swin-TNP also has the most accurate predictive uncertainties. These results are consistent with the findings from the previous two experiments. In addition, we find that for the Swin-TNP, the use of the multi PT-GE and multi KI-GE outperform the use of the single PT-GE and single KI-GE, respectively.

**Table 2:** Test log-likelihood ($\uparrow$) and RMSE ($\downarrow$) for the the multi-modal wind speed dataset. All grids have a grid size of $4$ in the first dimension (time). The standard errors of the log-likelihoods are all below $0.02$, and of the RMSE below $0.005$. FPT: forward pass time for a batch size of eight in ms. Params: number of model parameters in units of M.

| Model | GE | Grid size | Log-lik. $\uparrow$ | RMSE $\downarrow$ | FPT | Params |
|---|---|---|---|---|---|---|
| CNP | - | - | $-1.593$ | 2.536 | 33 | 0.66 |
| PT-TNP | - | $M = 64$ | 3.988 | 1.185 | 166 | 1.79 |
| ConvCNP | SetConv | $24 \times 60$ | 6.143 | 0.784 | 210 | 14.41 |
| ViTNP | multi KI-GE | $12 \times 30$ | 5.371 | 0.908 | 349 | 1.49 |
| ViTNP | multi PT-GE | $12 \times 30$ | 7.754 | 0.651 | 374 | 2.36 |
| ViTNP | multi KI-GE | $24 \times 60 \to 12 \times 30$ | 6.906 | 0.718 | 372 | 1.55 |
| ViTNP | multi PT-GE | $24 \times 60 \to 12 \times 30$ | 7.288 | 0.681 | 392 | 3.25 |
| Swin-TNP | multi KI-GE | $24 \times 60$ | 7.603 | 0.651 | 355 | 1.49 |
| Swin-TNP | multi PT-GE | $24 \times 60$ | **8.603** | **0.577** | 375 | 3.19 |
| Swin-TNP | single KI-GE | $24 \times 60$ | 7.794 | 0.642 | 366 | 1.39 |
| Swin-TNP | single PT-GE | $24 \times 60$ | 8.073 | 0.614 | 364 | 1.67 |
| ConvCNP | SetConv | $48 \times 120$ | 7.841 | 0.615 | $64^{10}$ | 14.41 |
| Swin-TNP | multi PT-GE | $48 \times 120$ | **9.383** | **0.509** | 369 | 6.51 |

## 5.4 Discussion of Results

We summarise our conclusions across all three experiments as follows: 1. gridded TNPs significantly outperform all baselines while maintaining relatively low computational complexity; 2. the PT-GE outperforms the KI-GE in almost all gridded TNPs, particularly when the grid is coarse; 3. when modelling multiple data sources, the use of source-specific grid encoders outperforms a single, unified grid encoder; 4. Swin-TNP achieves either comparable or better performance than the ViTNP at a smaller computational cost; and 5. in the grid decoder, nearest-neighbour cross-attention both reduces the computational cost and improves predictive performance relative to full cross-attention. In Appendix E, we provide additional experimental results showing that these conclusions are robust to changes in model size and dataset composition.

## 6 Conclusion

This paper introduces gridded TNPs, an extension to the family of TNPs which facilitates the use of efficient attention-based transformer architectures such as the ViT and Swin Transformer. Gridded TNPs decompose the computational backbone of TNPs into three distinct components: 1. the *grid encoder*, which moves point-wise data representations onto a structured grid of pseudo-tokens; 2. the *grid processor*, which processes this grid using computationally efficient operations; and 3. the *grid decoder*, which evaluates the processed grid at arbitrary input locations. In achieving this, we develop the pseudo-token grid encoder—a novel approach to moving unstructured spatio-temporal data onto a grid of pseudo-tokens—and the pseudo-token grid decoder—a computationally efficient approach of evaluating a grid of pseudo-tokens at arbitrary input locations. We compare the performance of a number of gridded TNPs against several strong baselines, demonstrating significantly better performance on large-scale synthetic and real-world regression tasks involving context sets containing over $100,000$ datapoints. This work marks an initial step towards building architectures for modelling large amounts of unstructured spatio-temporal observations. We believe that the methods developed in this paper can be used to both improve and broaden the capabilities of existing machine learning models for spatio-temporal data, including those targeting weather and environmental forecasting. We look forward to pursuing this in future work.

---

[10]The FPT for the larger ConvCNP is smaller than for the smaller ConvCNP as we were forced to use $k = 27$ nearest-neighbour grid decoding to avoid out of memory issues.

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

# A  A UNIFYING CONSTRUCTION OF CONDITIONAL NEURAL PROCESSES

The many variants of CNPs differ in their construction of the predictive distribution $p(\cdot|\mathbf{x}_t, \mathcal{D}_c)$. Here, we introduce a construction of CNPs that generalises all variants. CNPs are formed from three components: the *encoder*, the *processor*, and the *decoder*. The encoder, $e\colon \mathcal{X} \times \mathcal{Y} \to \mathcal{Z}$, first encodes each $(\mathbf{x}_{c,n}, \mathbf{y}_{c,n}) \in \mathcal{D}_c$ into some latent representation, or *token*, $\mathbf{z}_{c,n} \in \mathcal{Z}$. The processor, $\rho\colon \left( \bigcup_{n=0}^{\infty} \mathcal{Z}^n \right) \times \mathcal{X} \to \mathcal{Z}$, processes the set of context tokens $e(\mathcal{D}_c) = \{e(\mathbf{x}_{c,n}, \mathbf{y}_{c,n})\}_n$ together with the target input $\mathbf{x}_t$ to obtain a target dependent token, $\mathbf{z}_t \in \mathcal{Z}$.[11] Finally, the decoder, $d\colon \mathcal{Z} \to \mathcal{P}_\mathcal{Y}$, maps from the target token to the predictive distribution over the output at that target location. Here, $\mathcal{P}_\mathcal{Y}$ denotes the set of distributions over $\mathcal{Y}$. We illustrate this decomposition in Figure 5.

$$\{\mathbf{x}_{c,n}, \mathbf{y}_{c,n}\}_n \xrightarrow{\text{Encode, } e(\cdot)} \{\mathbf{z}_{c,n}\}_n \xrightarrow{\text{Process, } \rho(\cdot, \mathbf{x}_t)} \mathbf{z}_t \xrightarrow{\text{Decode, } d(\cdot)} p(\cdot \mid \mathcal{D}_c, \mathbf{x}_t)$$

**Figure 5:** A unifying construction of CNPs, with $\mathcal{D}_c = \{(\mathbf{x}_{c,n}, \mathbf{y}_{c,n})\}_n$ and $\mathbf{z}_{c,n} = e(\mathbf{x}_{c,n}, \mathbf{y}_{c,n})$.

We present below several schematics showing the architectures of different members of the CNP family, and detail how they can be constructed following this universal construction.

**Original CNP**  The first architecture is based on Garnelo et al. (2018a) and is the least complex of the CNP variants, using a summation as the permutation-invariant aggregation. The diagram is shown in Figure 6. The encoder of a CNP is an MLP, which maps from each concatenated pair $(\mathbf{x}_{c,n}, \mathbf{y}_{c,n}) \in \mathcal{D}_c$ to some representation $\mathbf{z}_{c,n} \in \mathbb{R}^{D_z}$. The processor sums together these representations, and combines the aggregated representation with the target input using $\tau$: $\rho(\{\mathbf{z}_{c,n}\}_n, \mathbf{x}_t) = \tau(\sum_n \mathbf{z}_{c,n}, \mathbf{x}_t)$. $\tau$ is often just the concatenation operation. Finally, the decoder consists of another MLP which maps from $\mathbf{z}_t = \tau(\sum_n \mathbf{z}_{c,n}, \mathbf{x}_t)$ to the parameter space of some distribution over the output space (e.g. Gaussian).

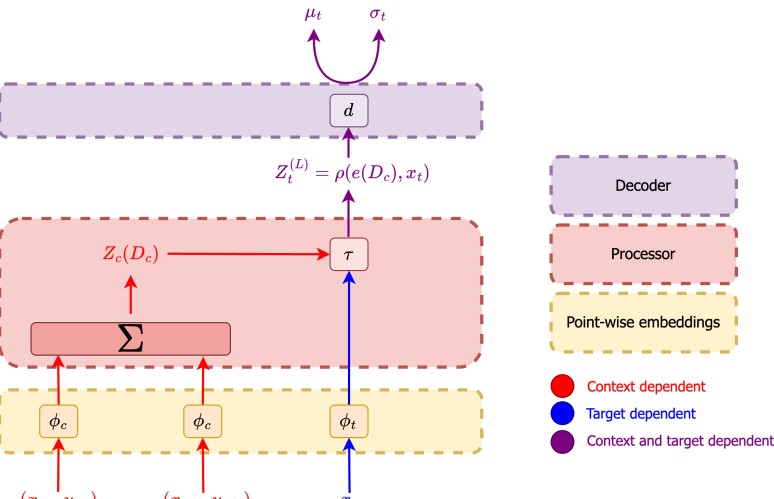

**Figure 6:** A diagram illustrating the architecture of the plain CNP (Garnelo et al., 2018a). First, the context set $(\mathbf{x}_{c,n}, \mathbf{y}_{c,n})$ and the target tokens $\mathbf{x}_{t,n}$ are encoded using point-wise embeddings. These are fed into the CNP processor, which performs a simple permutation-invariant aggregation of the context tokens. These are then concatenated with the target tokens and fed into the decoder, which outputs the parameters of the specified NP distribution based on the target representation (in this case, mean and variance of a Gaussian).

---

[11]The space of target dependent tokens does **not** need to be the same as that of context tokens—we have used $\mathcal{Z}$ in both cases for simplicity. It is also possible for $\mathcal{Z}$ to be the product of multiple spaces, e.g. $\mathcal{Z} = \mathcal{Z}_{token} \times \mathcal{X}$ where we retain information about the input locations.

**ConvCNP** Another member of the CNP family is the ConvCNP (Gordon et al., 2019), that embeds sets into function space in order to achieve translation equivariance. This can lead to more efficient training in applications where such an inductive bias is appropriate, such as stationary time-series or spatio-temporal regression tasks. We show in Figure 7 a schematic of the architecture. The encoder of a ConvCNP is simply the identity function. The processor then encodes the discrete function represented by input-output pairs $\{(\mathbf{x}_{c,n}, \mathbf{y}_{c,n})\}_n$ onto a regular grid using the kernel-interpolation grid encoder (KI-GE). It then processes this grid using a CNN, which is afterwards combined with the target location using a kernel-interpolation grid decoder (KI-GD). Letting $\mathbf{U} = \{\mathbf{u}_m\}_m$ denote the set of $M$ values on gridded locations $\mathbf{V} = \{\mathbf{v}_m\}_m$, we can decompose the ConvCNP processor as

$$\mathbf{u}_m \leftarrow \sum_n [1, \ \mathbf{y}_{c,n}]^T \psi_{ge}(\mathbf{v}_m - \mathbf{x}_{c,n}) \tag{6}$$

$$\mathbf{U} \leftarrow \mathrm{CNN}(\mathbf{U}, \mathbf{V}) \tag{7}$$

$$\mathbf{z}_t \leftarrow \sum_m \mathbf{u}_m \psi_{gd}(\mathbf{x}_t - \mathbf{v}_m). \tag{8}$$

Here, $\psi_{ge}, \ \psi_{gd} \colon \mathbb{R}^{D_x} \times \mathbb{R}^{D_x} \to \mathbb{R}$ denote the KI-GE kernel and KI-GD kernel. As with the CNP, the decoder is an MLP mapping to the parameter space of some distribution over the output space.

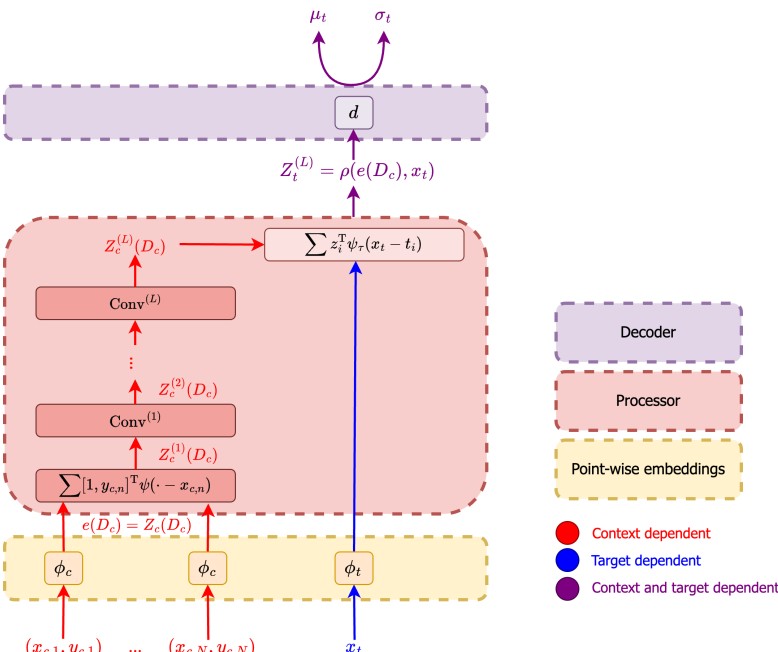

**Figure 7:** A diagram illustrating the architecture of the ConvCNP (Gordon et al., 2019). First, the context set $(\mathbf{x}_{c,n}, \mathbf{y}_{c,n})$ and the target tokens $\mathbf{x}_{t,n}$ are encoded using point-wise embeddings. These are fed into the ConvCNP processor, which uses a KI-GE to project the tokens into function space. These are then evaluated at discrete locations using a pre-specified resolution, followed by multiple layers of a CNN-based architecture acting upon the discretised signal. To decode at arbitrary locations, a KI-GD is used, giving rise to the target token representation. This is fed into the decoder, which outputs the parameters of the specified NP distribution (in this case, mean and variance of a Gaussian).

**TNPs** There are a number of different architectures used for the different members of the TNP family. We provide below diagrams for two members mentioned in the main paper, namely the TNP of Nguyen & Grover (2022) and the induced set transformer (ISTNP) of Lee et al. (2019). For the standard TNP, the encoder consists of an MLP mapping from each $(\mathbf{x}_{c,n}, \mathbf{y}_{c,n}) \in \mathcal{D}_c$ to some representation (token) $\mathbf{z}_{c,n} \in \mathbb{R}^{D_z}$. The processor begins by embedding the target location in

the same space as the context tokens, giving $\mathbf{z}_t \in \mathbb{R}^{D_z}$. It then iterates between applying MHSA operations on the set of context tokens, and MHCA operations from the set of context tokens into the target token:

$$\left.\begin{array}{l} \mathbf{Z}_c \leftarrow \text{MHSA}(\mathbf{Z}_c) \\ \mathbf{z}_t \leftarrow \text{MHCA}(\mathbf{z}_t; \ \mathbf{Z}_c) \end{array}\right\} \times L. \tag{9}$$

Again, the decoder consists of an MLP mapping from $\mathbf{z}_t$ to the parameter space of some distribution over outputs.

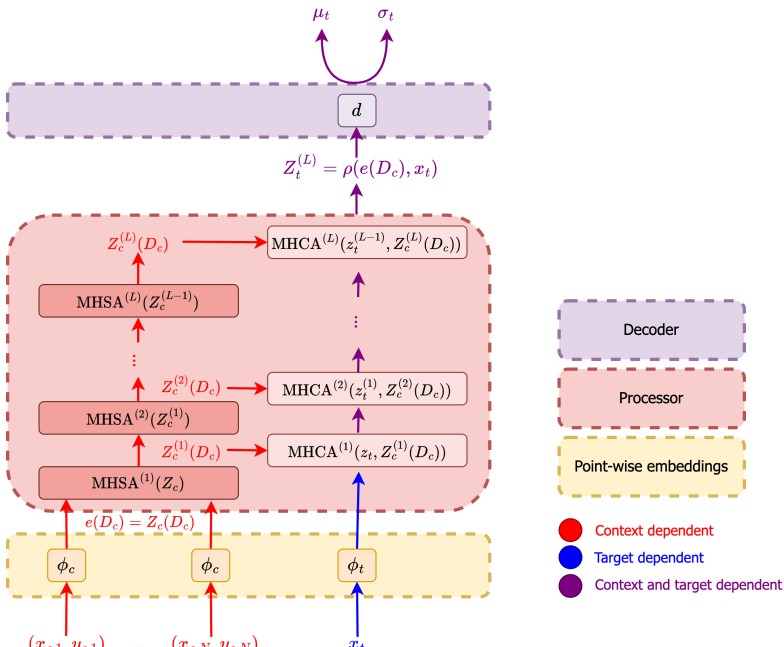

**Figure 8:** A diagram illustrating the architecture of the TNP (Nguyen & Grover, 2022). First, the context set $(\mathbf{x}_{c,n}, \mathbf{y}_{c,n})$ and the target tokens $\mathbf{x}_{t,n}$ are encoded using point-wise embeddings to obtain the context set representation $\mathbf{Z}_c$ and target representation $\mathbf{Z}_t$. These are fed into the TNP processor, which takes in the union of $[\mathbf{Z}_c, \mathbf{Z}_t]$ and outputs the token corresponding to the target inputs $\mathbf{Z}_t^{(L)}$. At each layer of the processor, the context set representation is first updated through an MHSA layer, which is then used to modulate the target set representation through a MHCA layer between the target set representation from the previous layer and the updated context representation. Finally, the decoder outputs the parameters of the specified NP distribution based on the target representation from the final layer (in this case, mean and variance of a Gaussian).

One of the main limitations of TNPs is the cost of the attention mechanism, which scales quadratically with the number of input tokens. Several works (Feng et al., 2023; Lee et al., 2019) addressed this shortcoming by incorporating ideas from the Perceiver-style architecture (Jaegle et al., 2021) into NPs. The strategy is to introduce a set of $M$ 'pseudo-tokens' which act as an information bottleneck between the context and target sets. Provided that $M << N_c$, where $N_c$ is the number of context points, this leads to a significant reduction in computational complexity. The architecture we consider in this work is called the induced set transformer NP (ISTNP), and differs from the plain TNP in the calculations performed in the processor. At each layer, the pseudo-token representation is first updated through an MHCA operation from the context set to the pseudo-tokens. The updated representation is then used to modulate the context and target sets separately, through separate MHCA operations:

$$\left.\begin{array}{l} \mathbf{U} \leftarrow \text{MHCA}(\mathbf{U}; \ \mathbf{Z}_c) \\ \mathbf{z}_t \leftarrow \text{MHCA}(\mathbf{z}_t; \ \mathbf{U}) \\ \mathbf{Z}_c \leftarrow \text{MHCA}(\mathbf{Z}_c; \ \mathbf{U}) \end{array}\right\} \times L. \tag{10}$$

Thus, as apparent in Figure 9, the context and target sets never interact directly, but only through the 'pseudo-tokens'. The computational cost at each layer reduces from $\mathcal{O}(N_c^2 + N_c N_t)$ in the plain TNP, where $N_c$ and $N_t$ represent the number of context and target points, respectively, to $\mathcal{O}(M(2N_c + N_t))$. This is a significant reduction provided that $M << N_c$, resulting in an apparent linear dependency on $N_c$. However, in practice, $M$ is not independent of $N_c$, with more pseudo-tokens needed as the size of context set increases.

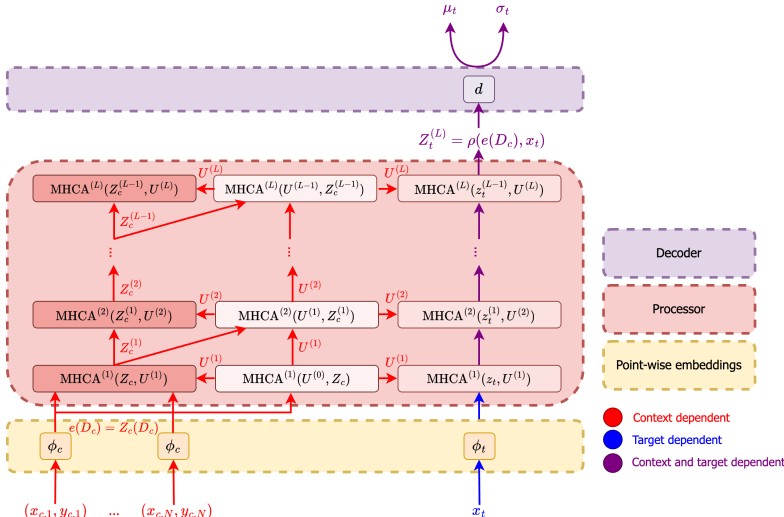

**Figure 9:** A diagram illustrating the architecture of the ISTNP (Lee et al., 2019). As opposed to the regular TNP of Nguyen & Grover (2022), the ISTNP uses a 'summarised' representation of the context set through the use of pseudo-tokens. They are first randomly initialised ($\mathbf{U}^{(0)}$). Then, their representation is updated through cross attention with the context set representation from the previous layer (i.e. at layer $l$: $\mathbf{U}^{(l)} = \text{MHCA}^{(l)}(\mathbf{U}^{(l-1)}, \mathbf{Z}_c^{(l)})$). This updated set of pseudo-tokens $\mathbf{U}^{(l)}$ is then used to modulate both the context set representation at the current layer through cross-attention (i.e. $\mathbf{Z}_c^{(l)} = \text{MHCA}(\mathbf{Z}_c^{(l-1)}, \mathbf{U}^{(l)})$), as well as the target set representation (i.e. $\mathbf{Z}_t^{(l)} = \text{MHCA}(\mathbf{Z}_t^{(l-1)}, \mathbf{U}^{(l)})$). Thus, the context and target set representations do not interact directly, but only through the pseudo-tokens, which act as a bottleneck of information flow between the two in order to decrease the computational demands of the plain TNP.

## B  KERNEL-INTERPOLATION GRID ENCODER

Let $\mathbf{z}_n \in \mathbb{R}^{D_z}$ denote the token representation of input-output pair $(\mathbf{x}_n, \mathbf{y}_n)$ after point-wise embedding. We introduce the set of grid locations $\mathbf{V} \in \mathbb{R}^{M_1 \times \cdots M_{D_x} \times D_x}$. For ease of reading, we shall replace the product $\prod_{d=1}^{D_x} M_d$ with $M$ and the indexing notation $m_1, \ldots, m_{D_x}$ with $m$.

The kernel-interpolation grid encoder obtains a pseudo-token representation $\mathbf{U} \in \mathbb{R}^{M \times D_z}$ of $\mathcal{D}_c$ on the grid $\mathbf{V}$ by interpolating from *all* tokens $\{\mathbf{z}_{c,n}\}_n$ at corresponding locations $\{\mathbf{x}_{c,n}\}_n$ to *all* pseudo-token locations:

$$\mathbf{u}_m \leftarrow \sum_n \mathbf{z}_{c,n} \psi(\mathbf{v}_m, \mathbf{x}_{c,n}) \quad \forall m \in M. \tag{11}$$

Here, $\psi \colon \mathcal{X} \times \mathcal{X} \to \mathbb{R}$ is the kernel used for interpolation, which we take to be the squared-exponential (SE) kernel when $\mathcal{X} = \mathbb{R}^{D_x}$:

$$\psi_{SE}(\mathbf{v}_m, \mathbf{x}_{c,n}) = \exp\left(-\sum_{d=1}^{D_x} \frac{(x_{c,n,d} - v_{m,d})^2}{\ell_d^2}\right) \tag{12}$$

where $\ell_d$ denotes the 'lengthscale' for dimension $d$. Similar to the pseudo-token grid encoder, we can restrict the kernel-interpolation grid encoder to interpolate only from sets of token $\{\mathbf{z}_{c,n}\}_{n \in \mathfrak{N}(\mathbf{v}_m; k)}$

for which $\mathbf{v}_m$ is amongst the $k$ nearest grid locations. The computational complexity of the kernel-interpolation and pseudo-token grid encoders differ only by a scale factor when this approach is used.

## C  NEAREST-NEIGHBOUR CROSS-ATTENTION IN THE GRID DECODER

In this section we provide more details, alongside schematics, of our nearest-neighbour cross-attention scheme. As mentioned in the main text, in the grid decoder we only allow a subset of the gridded pseudo-tokens to attend to the target token (those in the vicinity of it). Finding the nearest neighbours is not done in the standard k-nearest neighbours fashion, because we use the same number of nearest neighbours along each dimension of the original data (i.e. latitude and longitude for spatial interpolation; latitude, longitude and time for spatio-temporal interpolation). This is to ensure that we do not introduce specific preferences for any dimension. In the case the data lies on a grid with the same spacing in all its dimensions, the procedure becomes equivalent to k-nearest neighbours.

In practice, we specify the total number of nearest-neighbours we want to use $k$. We then compute the number of nearest-neighbours in each dimension by $k_{\text{dim}} = \text{ceil}(k^{\frac{1}{\dim(x)}})$, where $\dim(x)$ represents the dimensionality of the input. For efficient batching purposes, we tend to choose $k = (2n - 1)^{\dim(x)}$, where $n \in \mathbb{N}$ (i.e. 9 for experiments with latitude-longitude grids, 27 for experiments with latitude-longitude-time grids). We then find the indices in each dimension of these nearest-neighbours by performing an efficient search that leverages the gridded nature of the data, leading to a computational complexity of $\mathcal{O}(kN_t)$, with $N_t$ the number of target points. When the neighbours go off the grid (i.e. for targets very close to the edges of the grid), we only consider the number of viable (i.e. within the bounds of the grid) neighbours.

**Example in 2D**  This procedure is visualised in Figure 10, where we consider both grids with the same spacing along each dimension, as well as grids with different spacings. We cover both the case of a central target point, as well as a target point closer to the edges of the grid.

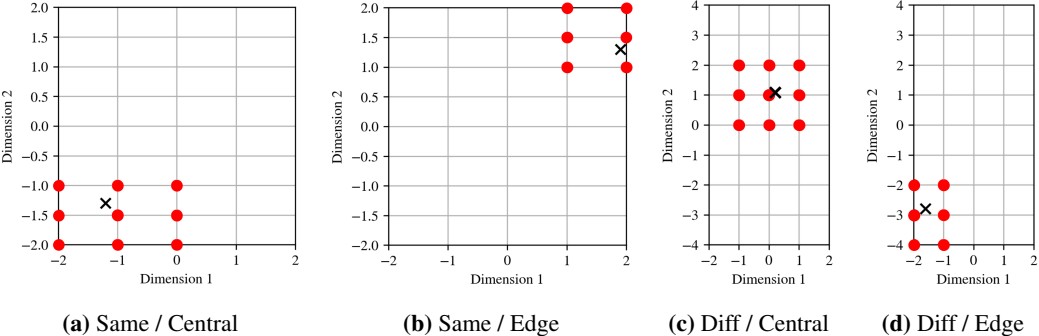

(a) Same / Central  (b) Same / Edge  (c) Diff / Central  (d) Diff / Edge

**Figure 10:** Example of our nearest-neighbours procedure in 2D on a $5 \times 9$ grid for 9 nearest neighbours. We consider four different cases. Same / Diff refers to whether the grid spacing is the same in the two dimensions or different. Central / Edge refers to the position of the target. In the case of an edge target, we do not consider invalid neighbours (i.e. those that are outside the grid bounds).

**Accounting for non-Euclidean geometry**  A lot of our experiments are performed on environmental data, distributed across the Earth. For our purposes, we assume the Earth shows cylindrical geometry, whereby there is no such thing as a grid edge along the longitudinal direction (i.e. a longitude of -180° is the same as 180°). This is not the same for latitude, where one extreme corresponds to the North Pole, and the other one to the South Pole. Thus, we would like to allow for the grid to 'roll' around the longitudinal direction when computing the nearest neighbours. In this case, there should be no edge target points along the longitudinal dimension. This procedure is graphically depicted in Figure 11 and we use it in our experiments on environmental data.

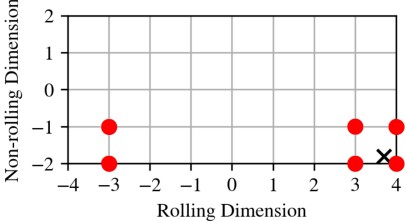

**Figure 11:** Example of nearest-neighbours procedure in 2D on a $5 \times 9$ grid for 9 nearest neighbours. We allow rolling along the horizontal dimension (e.g. longitude), but do not allow rolling along the vertical one (i.e. latitude). Hence, the neighbours extend on the other side of the grid horizontally, but not vertically. This example corresponds to cylindrical symmetry.

**Example for 3D data**   We also provide examples of the nearest-neighbours procedure on 3D spaces. These dimensions could represent, for example, latitude, longitude and time, or latitude, longitude and height/pressure levels. In Figure 12 we consider a case where we do not allow for rolling along any dimension, while in Figure 13 we allow for rolling along one of the dimensions.

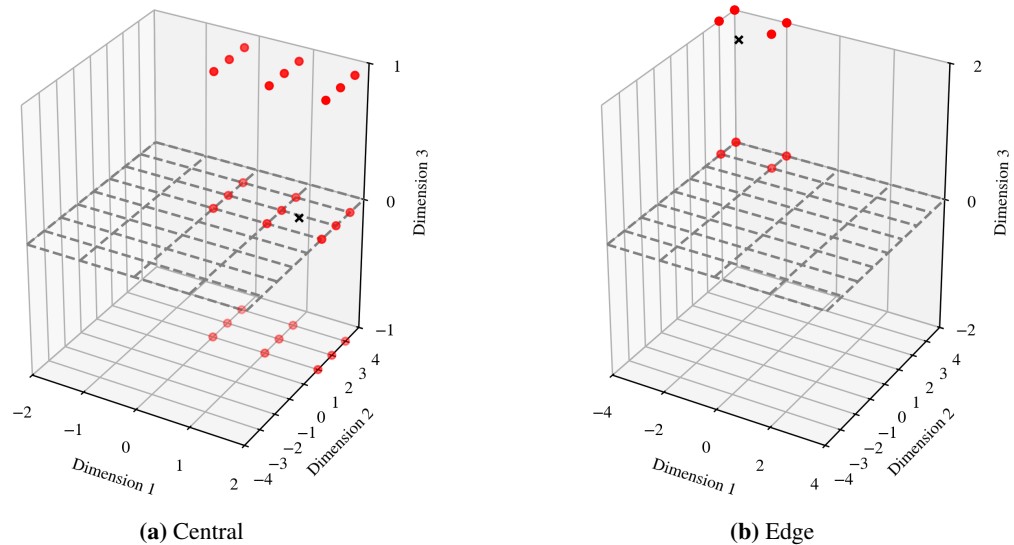

**(a)** Central                                         **(b)** Edge

**Figure 12:** Example of our nearest-neighbours procedure in 3D on a $5 \times 9 \times 3$ grid for 27 nearest neighbours. We consider two different cases—whether the target is central or near the edge of the grid. In the case of an edge target (right), we do not consider invalid neighbours (that are outside the grid bounds).

## D   HARDWARE SPECIFICATIONS

For the smaller synthetic GP regression experiment, we perform training and inference for all models on a single NVIDIA GeForce RTX 2080 Ti GPU with 20 CPU cores. For the other two, larger experiments, we perform training and inference for all models on a single NVIDIA A100 80GB GPU with 32 CPU cores.

## E   EXPERIMENT DETAILS

**Common optimiser details**   For all experiments and all models, we use the AdamW optimiser (Loshchilov & Hutter, 2017) with a fixed learning rate of $5 \times 10^{-4}$ and apply gradient clipping to gradients with magnitude greater than 0.5.

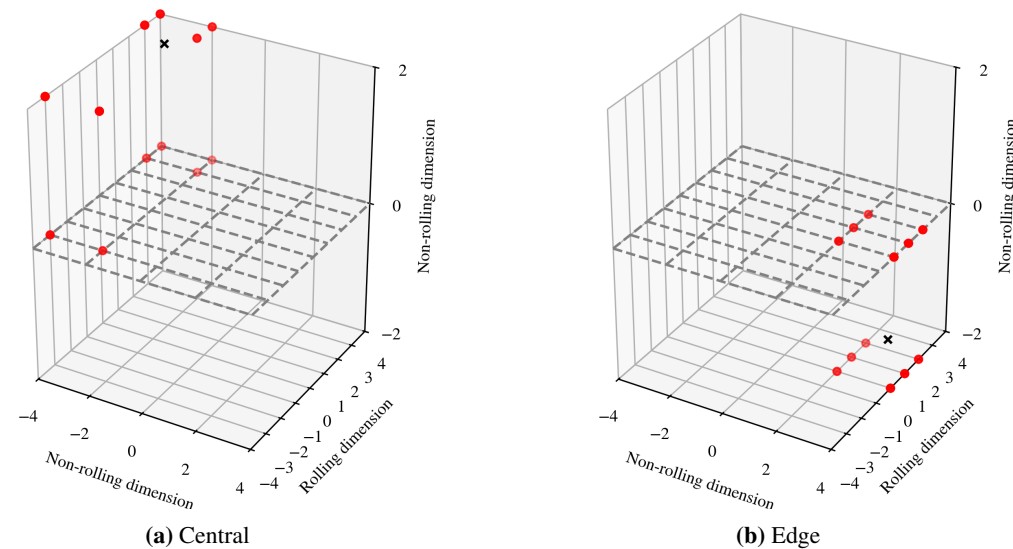

**(a)** Central            **(b)** Edge

**Figure 13:** Example of our nearest-neighbours procedure in 3D on a $5 \times 9 \times 3$ grid for 27 nearest neighbours. We allow for rolling along the second dimension, but not along any of the other ones.

**Common likelihood details** For all experiments and all models, we employ a Gaussian likelihood parameterised by a mean and inverse-softplus variance, i.e. the decoder of each model outputs

$$\boldsymbol{\mu}_t, \; \log(\exp \boldsymbol{\sigma}_t^2 - 1) = d(\mathbf{z}_t), \quad p(\,\cdot\mid \mathcal{D}_c, \mathbf{x}_t) = \mathcal{N}(\cdot; \boldsymbol{\mu}_t, \boldsymbol{\sigma}_t^2). \tag{13}$$

For the experiment modelling skin and 2m temperature (`skt` and `t2m`) with a richer context, we set a minimum noise level of $\boldsymbol{\sigma}_{\min}^2 = 0.01$ by parameterising

$$\boldsymbol{\mu}_t, \; \log(\exp(\boldsymbol{\sigma}_t - \boldsymbol{\sigma}_{\min})^2 - 1) = d(\mathbf{z}_t), \quad p(\,\cdot\mid \mathcal{D}_c, \mathbf{x}_t) = \mathcal{N}(\cdot; \boldsymbol{\mu}_t, \boldsymbol{\sigma}_t^2). \tag{14}$$

**CNP details** For the CNPs, we encode each $(\mathbf{x}_{c,n}, \mathbf{y}_{c,n}) \in \mathcal{D}_c$ in $\mathbb{R}^{D_z}$ using an MLP with two-hidden layers of dimension $D_z$. We obtain a representation for the entire context set by summing these representations together, $\mathbf{z}_c = \sum_n \mathbf{z}_{c,n}$, which is then concatenated with the target input $\mathbf{x}_t$. The concatenation $[\mathbf{z}_c, \mathbf{x}_t]$ is decoded using an MLP with two-hidden layers of dimension $D_z$. We use $D_z = 128$ in all experiments.

**ConvCNP details** For the ConvCNP model, we use a U-Net architecture (Ronneberger et al., 2015) for the CNN consisting of 11 layers with input size $C$. Between the five downward layers we apply pooling with size two. For the five upward layers, we use $2C$ input channels and $C$ output channels, as the input channels are formed from the output of the previous layer concatenated with the output of the corresponding downward layer. Between the upward layers we apply linear up-sampling to match the grid size of the downward layer. In all experiments, we use $C = 128$, a kernel size of five or nine, and a stride of one. We use SE kernels for the SetConv encoder and SetConv decoder with learnable lengthscales for each input dimension. The grid encoding is modified similarly to the pseudo-token grid encoder, whereby we only interpolate from the set of observations for which each grid point is the closest grid point. Unless otherwise specified, we also modify the grid decoding similarly to the pseudo-token grid decoder, whereby we only interpolate from the $k = 3^{D_x}$ nearest points on a distance-normalised grid to each target location. We resize the output of the SetConv encoder to dimension $C$ using an MLP with two hidden layers of dimension $C$. We resize the output of the SetConv decoder using an MLP with two hidden layers of dimension $C$.

**Common transformer details** For each MHSA / MHCA operation, we construct a layer consisting of two residual connections, two layer norm operations, one MLP, together with the MHSA /

MHCA operation as follows:

$$\widetilde{\mathbf{Z}} \leftarrow \mathbf{Z} + \text{MHSA/MHCA}(\text{layer-norm}_1(\mathbf{Z}))$$
$$\mathbf{Z} \leftarrow \widetilde{\mathbf{Z}} + \text{MLP}(\text{layer-norm}_2(\widetilde{\mathbf{Z}})). \tag{15}$$

All MHSA / MHCA operations use $H = 8$ heads, each with $D_V = 16$ dimensions. We use a $D_z = D_{QK} = 128$ throughout.

**PT-TNP details** We use an induced set transformer (IST) architecture for the PT-TNPs, with each layer consisting of the following set of operations:

$$\mathbf{U} \leftarrow \text{MHCA-layer}(\mathbf{U}; \mathbf{Z}_c)$$
$$\mathbf{z}_t \leftarrow \text{MHCA-layer}(\mathbf{z}_t; \mathbf{U}) \tag{16}$$
$$\mathbf{Z}_c \leftarrow \text{MHCA-layer}(\mathbf{Z}_c; \mathbf{U}).$$

In all experiments we use five layers, and encoder / decoder MLPs consisting of two hidden layers of dimension $D_z$.

**ViT details** The ViT architecture consists of optional patch encoding, followed by five MHSA layers. The patch encoding is implemented using a single linear layer. In all experiments we use five layers, and encoder / decoder MLPs consisting of two hidden layers of dimension $D_z$.

**Swin Transformer details** Each layer of the Swin Transformer consists of two MHSA layers applied to each window, and a shifting operation between them. Unless otherwise specified, we use a window size of four and shift size of two for all dimensions (except for the time dimension in the final experiment, as the original grid only has four elements in the time dimension). For the second experiment in which the grid covers the entire globe, we allow the Swin attention masks to 'roll' over the longitudinal dimension, allowing the pseudo-tokens near $180°$ longitude to attend to those near $-180°$ longitude. In all experiments, we use five Swin Transformer layers (10 MHSA layers in total), and encoder / decoder MLPs consisting of two hidden layers of dimension $D_z$. We found that the use of a hierarchical Swin Transformer—as used in the original Swin Transformer and models such as Aurora (Bodnar et al., 2024)—did not lead to any improvement in performance.

**Spherical harmonic embeddings** When modelling input data on the sphere (i.e. the final two experiments), the CNP, PT-TNP, and gridded TNP models first encode the latitude / longitude coordinates using spherical harmonic embeddings following Rußwurm et al. (2024) using 10 Legendre polynomials. We found this to improve performance in all cases.

**Temporal Fourier embeddings** When modelling input data through time (i.e. the final experiment), the CNP, PT-TNP, and gridded TNP models first encode the temporal coordinates using a Fourier embedding. The time value is originally provided in hours since 1st January 1970. Following Bodnar et al. (2024), we embed this using a Fourier embedding of the following form:

$$\text{Emb}(t) = \left[ \cos \frac{2\pi t}{\lambda_i}, \ \sin \frac{2\pi t}{\lambda_i} \right] \text{ for } 0 \le i < L/2. \tag{17}$$

where the $\lambda_i$ are log-spaced values between the minimum and maximum wavelength. We set $\lambda_{min} = 1$ and $\lambda_{max} = 8760$, the number of hours in a year. We use $L = 10$.

**Great-circle distance** For methods using the kernel-interpolation grid encoder (i.e. the ConvCNP and some gridded TNPs), we use the great-circle distance, rather than Euclidean distance, as the input into the kernel when modelling input data on the sphere. The haversine formula determines the great-circle distance between two points $\mathbf{x}_1 = (\varphi_1, \lambda_2)$ and $\mathbf{x}_2 = (\varphi_2, \lambda_2)$, where $\lambda$ and $\varphi$ denote the latitude and longitude, and is given by:

$$d(\mathbf{x}_1, \mathbf{x}_2) = 2r \arcsin \left( \sqrt{\frac{1 - \cos(\Delta\varphi) + \cos\varphi_1 \cdot \cos\varphi_2 \cdot (1 - \cos(\Delta\lambda))}{2}} \right) \tag{18}$$

where $\Delta\varphi = \varphi_2 - \varphi_1$, $\Delta\lambda = \lambda_2 - \lambda_1$ and $r$ is taken to be 1.

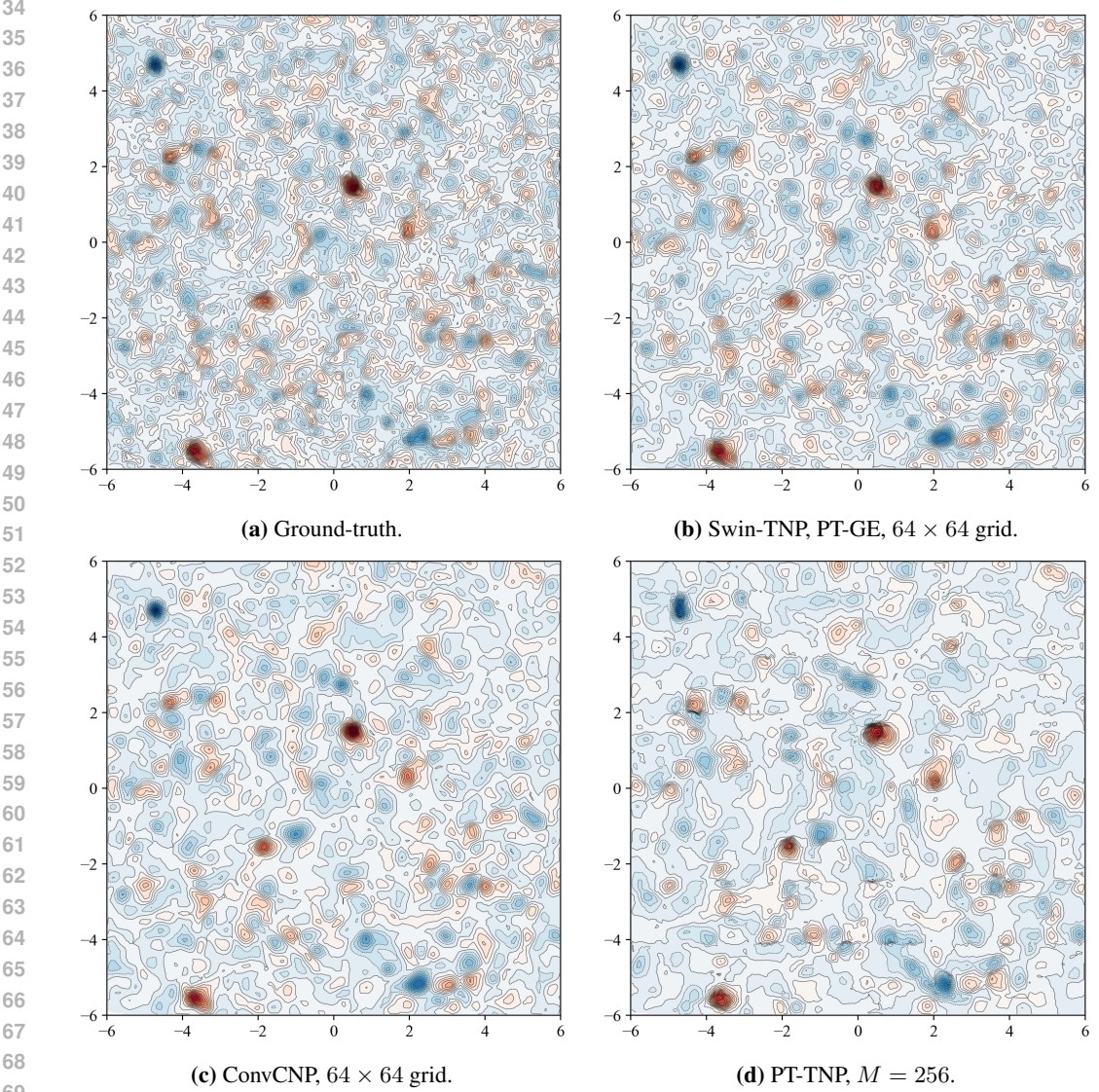

**(a)** Ground-truth.

**(b)** Swin-TNP, PT-GE, $64 \times 64$ grid.

**(c)** ConvCNP, $64 \times 64$ grid.

**(d)** PT-TNP, $M = 256$.

**Figure 14:** A comparison between the predictive means of a selection of CNP models on a synthetic GP dataset with $\ell = 0.1$. The noiseless ground-truth dataset is shown in Figure 14a, and the context set is a randomly sampled set of $N_c = 1 \times 10^4$ noisy observations of this. The colour corresponds to the output value, with the same scale used in each plot. Observe the complexity of the ground-truth dataset, which the Swin-TNP's predictive mean resembles. The ConvCNP and PT-TNP's predictive means are notably smoother.

### E.1    META-LEARNING GAUSSIAN PROCESS REGRESSION

We utilise the `GPyTorch` software package (Gardner et al., 2018) for generating synthetic samples from a GP. As the number of datapoints in each sampled dataset is very large by GP standards ($1.1 \times 10^4$), we approximate the SE kernel using structured kernel interpolation (SKI) (Wilson & Nickisch, 2015) with 100 grid points in each dimension. We use an observation noise of $\sigma_n = 0.1$ for the smaller and larger lengthscale tasks. In Figure 14, we show an example dataset generated using a lengthscale of $0.1$ to demonstrate the complexity of these datasets. We were unable to compute ground truth log-likelihood values for these datasets without running into numerical issues.

In addition, we also plot the predictive means (Figure 15) and predictive errors (Figure 16) in the form of heatmaps for a number of CNP models on a different example dataset.

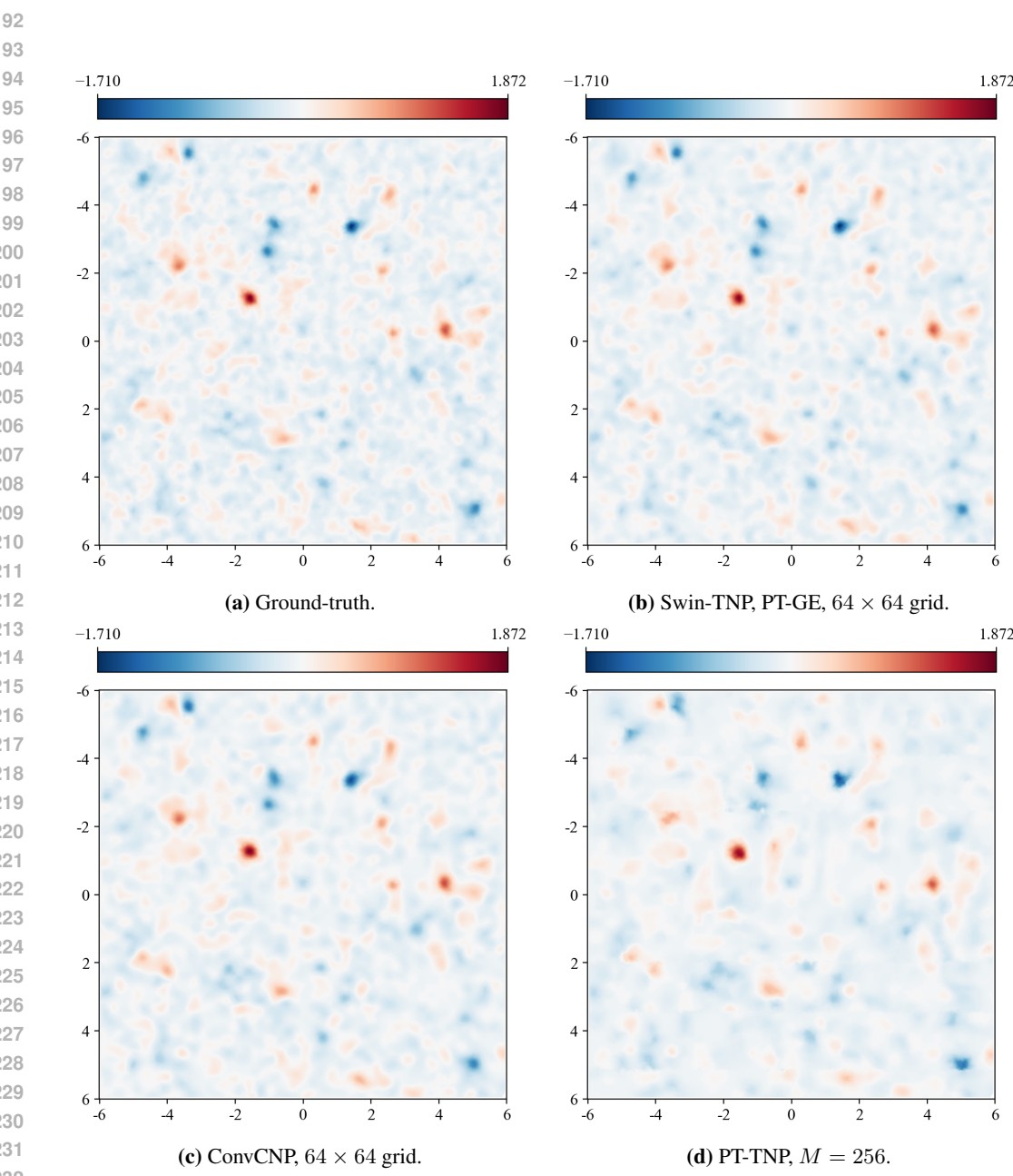

**Figure 15:** A comparison between the predictive means of a selection of CNP models on a synthetic GP dataset with $\ell = 0.1$. The noiseless ground-truth dataset is shown in Figure 15a, and the context set is a randomly sampled set of $N_c = 1 \times 10^4$ noisy observations of this. The colour corresponds to the output value, with the same scale used in each plot.

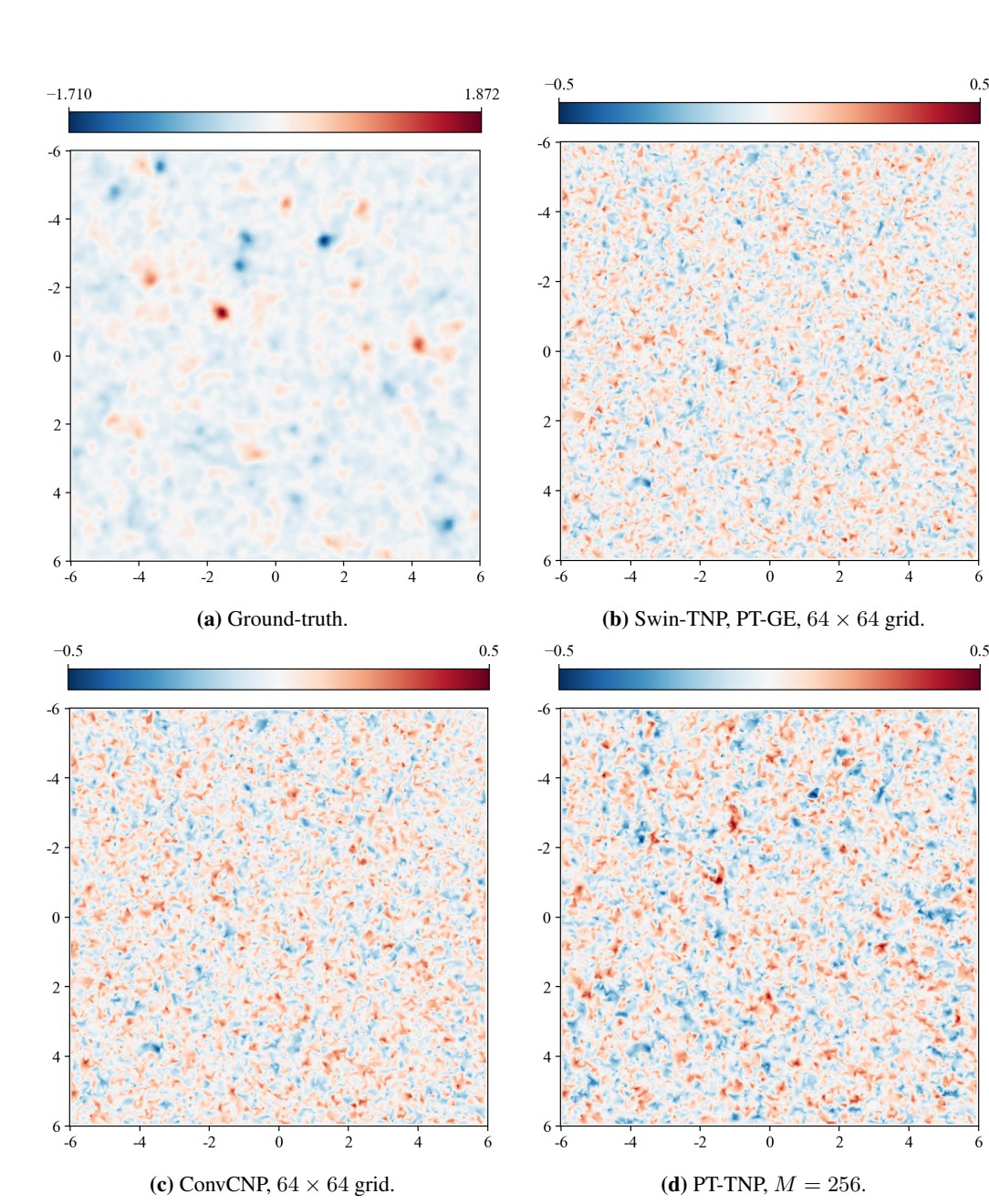

**Figure 16:** A comparison between the difference between the predictive mean and ground-truth for a selection of CNP models on a synthetic GP dataset with $\ell = 0.1$. The noiseless ground-truth dataset is shown in Figure 16a, and the context set is a randomly sampled set of $N_c = 1 \times 10^4$ noisy observations of this. The colour corresponds to the prediction error, with the same scale used in each plot.

We train all models for $500,000$ iterations on $160,000$ pre-generated datasets using a batch size of eight. For all models, excluding the ConvCNP, we apply Fourier embeddings to each input dimension with $L = 64$ wavelengths with $\lambda_{min} = 0.01$ and $\lambda_{max} = 12$. We found this to significantly improve the performance of all models. In Table 3, we provide test log-likelihood values for a number of gridded TNPs and baselines for both tasks. We observe that even when increasing the size of the baseline models they still underperform the smaller gridded TNPs. We include results for the Swin-TNP with full attention grid decoding (no NN-CA), which fail to model the more complex dataset when using the PT-GE.

**Table 3:** Test log-likelihood ($\uparrow$) for the synthetic GP regression dataset. FPT: forward pass time for a batch size of eight in ms. Params: number of model parameters in units of M.

| Model | Grid encoder | Grid size | $\ell = 0.5$ ($\uparrow$) | $\ell = 0.1$ ($\uparrow$) | FPT | Params |
|---|---|---|---|---|---|---|
| CNP | - | - | $-0.406$ | $0.112$ | 9 | 0.21 |
| PT-TNP | - | $M = 128$ | $0.819$ | $0.558$ | 53 | 1.50 |
| PT-TNP | - | $M = 256$ | $0.819$ | $0.565$ | 74 | 1.52 |
| ConvCNP | SetConv | $32 \times 32$ | $0.801$ | $0.536$ | 13 | 2.11 |
| ConvCNP | SetConv | $64 \times 64$ | $0.830$ | $0.681$ | 93 | 6.70 |
| ViTNP | KI-GE | $32 \times 32 \to 16 \times 16$ | $0.841$ | $0.722$ | 30 | 1.16 |
| ViTNP | PT-GE | $32 \times 32 \to 16 \times 16$ | $0.841$ | $0.721$ | 32 | 1.39 |
| ViTNP | KI-GE | $16 \times 16$ | $0.833$ | $0.711$ | 28 | 1.09 |
| ViTNP | PT-GE | $16 \times 16$ | $0.840$ | $0.712$ | 29 | 1.22 |
| ViTNP | KI-GE | $64 \times 64 \to 32 \times 32$ | $0.842$ | $0.728$ | 44 | 1.16 |
| ViTNP | PT-GE | $64 \times 64 \to 32 \times 32$ | $0.836$ | $0.727$ | 56 | 1.78 |
| ViTNP | KI-GE | $32 \times 32$ | $0.830$ | $0.725$ | 47 | 1.09 |
| ViTNP | PT-GE | $32 \times 32$ | $0.837$ | $0.728$ | 53 | 1.32 |
| Swin-TNP | KI-GE | $32 \times 32$ | $0.844$ | $0.723$ | 39 | 1.09 |
| Swin-TNP | PT-GE | $32 \times 32$ | $0.844$ | $0.723$ | 42 | 1.32 |
| Swin-TNP | Avg-GE | $32 \times 32$ | $0.840$ | $0.723$ | 34 | 1.22 |
| Swin-TNP | KI-GE | $64 \times 64$ | $0.846$ | $0.728$ | 62 | 1.09 |
| Swin-TNP | PT-GE | $64 \times 64$ | $\mathbf{0.847}$ | $\mathbf{0.730}$ | 69 | 1.72 |
| Swin-TNP | Avg-GE | $64 \times 64$ | $0.845$ | $0.725$ | 58 | 1.62 |
| Swin-TNP (no NN-CA) | KI-GE | $32 \times 32$ | $0.834$ | $0.716$ | 45 | 1.09 |
| Swin-TNP (no NN-CA) | PT-GE | $32 \times 32$ | $0.837$ | $0.109$ | 48 | 1.32 |

**ConvCNP** For the ConvCNP models, we use a regular CNN architecture with $C = 128$ channels and five layers. We use a kernel size of five for the smaller ConvCNP ($32 \times 32$ grid) and a kernel size of nine for the larger ConvCNP ($64 \times 64$).

**Swin-TNP** For the Swin-TNP models, we use a window size of $4 \times 4$ for the smaller model ($32 \times 32$ grid) and a window size of $8 \times 8$ for the larger model ($64 \times 64$ grid). The shift size is half the window size in each case. We also provide results when a simple average pooling is used for the grid encoder, which is similar to Xu et al. 2024 except that the pooling is performed in token space rather than on raw observations.

### E.1.1 SMALL-SCALE META-LEARNING GAUSSIAN PROCESS REGRESSION

We also consider a smaller GP regression task with datasets drawn from a GP with SE kernel with lengthscale $\ell = 0.1$. Each dataset in this smaller task has a randomly sized context set, $N_c \sim \mathcal{U}\{1, 1000\}$, and a fixed sized target set $N_t = 100$. The inputs are sampled uniformly in the range $[-2, 2]$ in each dimension. The use of a smaller dataset allows us to make comparisons with the regular TNP. In Table 4, we compare the performance of the best performing smaller ViTNP and Swin-TNP gridded TNPs from the paper with the regular TNP, implemented with five MHSA layers, token dimension $D_z = 128$, $H = 8$ heads and $D_Q = D_{KV} = 16$. We include the standard error of the mean test log-likelihood, which demonstrate that there is no significant difference in performance between the regular TNP and Swin-TNP. It should be noted, however, that the regular TNP is more computationally efficient than both gridded TNPs for this small-scale dataset. This reflects the

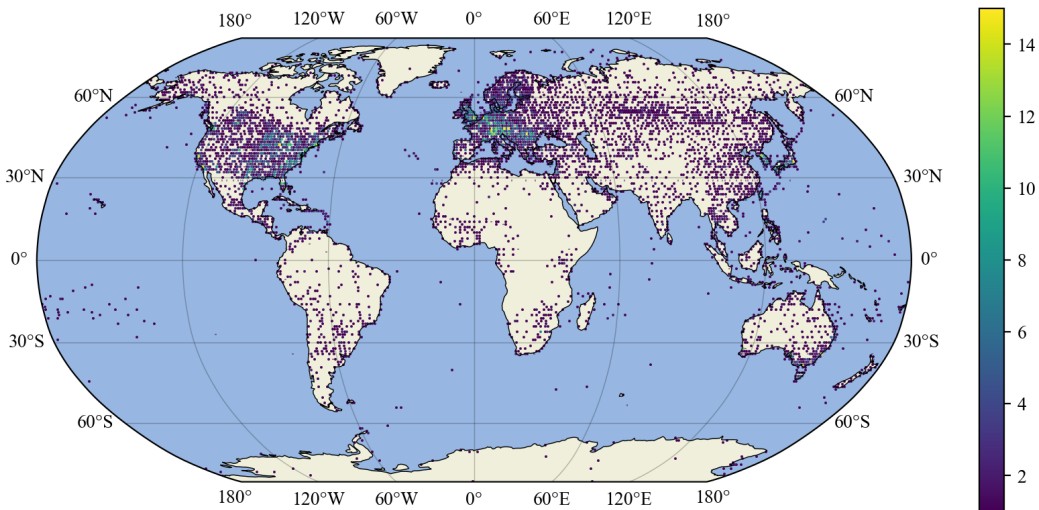

**Figure 17:** Station distribution within $1° \times 1°$ patches. The colour indicates the number of stations within each patch, clipped from a maximum value of 22 to 15. The distribution is far from uniform, with dense station areas in continents such as North America, Europe and parts of Asia, and a sparser distribution in Africa, South America, and in the oceans.

suitability of gridded TNPs for large-scale datasets, as there is little difference in forward pass time for the small-scale datasets here and the large-scale datasets considered in the paper for the gridded TNPs. In contrast, the TNP cannot be implemented on the large-scale datasets considered in the paper due to the quadratic computational and memory complexity associated with full attention.

**Table 4:** Test log-likelihood ($\uparrow$) for the synthetic GP regression dataset. FPT: forward pass time for a batch size of eight in ms. Params: number of model parameters in units of M.

| Model | Grid encoder | Grid size | Test log-likelihood ($\uparrow$) | FPT | Params |
|---|---|---|---|---|---|
| TNP | - | - | $\mathbf{-0.596 \pm 0.02}$ | 17 | 0.60 |
| ViTNP | PT-GE | $32 \times 32 \to 16 \times 16$ | $-0.657 \pm 0.02$ | 22 | 1.39 |
| Swin-TNP | PT-GE | $32 \times 32$ | $\mathbf{-0.616 \pm 0.02}$ | 33 | 1.32 |

### E.2 Combining Weather Station Observations with Structured Reanalysis

Inspired by the real-life assimilation of 2m temperature (t2m), we use the ERA5 reanalysis dataset to extract skin temperature (skt) and 2m temperature (t2m) at a $0.25°$ resolution (corresponding to a $721 \times 1440$ grid). We then coarsen the skt grid to a $180 \times 360$ grid, corresponding to $1°$ in both the latitudinal and longitudinal directions. This implies that, because t2m lies on a finer grid, it essentially becomes an off-the-grid variable with respect to the coarsened grid on which skt lies. In order for the experimental setup to better reflect real-life assimilation conditions, we assume to only observe off-the-grid t2m values at real weather station locations[12]. In total, there are $9,957$ such weather station locations, extracted from the HadISD dataset (Dunn et al., 2012). We show their geographical location in Figure 17.

For each task, we first randomly sample a time point, and then use the entire coarsened skt grid as the on-the-grid context data ($64,800$ points), as well as $N_{\text{off},c}$ off-the-grid t2m context points randomly sampled from the station locations. In the experiment from the main paper, $N_{\text{off},c} \sim \mathcal{U}_{[0,0.3]}$, but we also consider the case of richer off-the-grid context sets with $N_{\text{off},c} \sim \mathcal{U}_{[0.25,0.5]}$ in Table 8. The target locations are all the $9,957$ station locations.

---

[12]More specifically, because the 2m temperature values come from the gridded ERA5 data, we only consider the nearest grid points to the true station locations as valid off-the-grid locations (i.e. if a station has coordinates at $(44.19°, 115.43°)$ latitude-longitude, we consider the grid point at $(44.25°, 115.5°)$)

We train all models for $300,000$ iterations on the hourly data between $2009 - 2017$ with a batch size of eight. Validation is performed on 2018 and testing on 2019. The test metrics are reported for $16,000$ data samples. Experiment specific architecture choices are described below.

**Input embedding**   We use spherical harmonic embeddings for the latitude / longitude values. These are not used in the ConvCNP model as the ConvCNP does not modify the inputs in order to maintain translation equivariance (in this case, with respect to the great-circle distance).

**Grid sizes**   For the main experiment (with results reported in Table 1), we chose a grid size of $64 \times 128$ for the ConvCNP and Swin-TNP models, corresponding to a grid spacing of $2.8125°$ in both the latitudinal and longitudinal directions.

In Table 8 we report results for a richer context set using a grid size of $128 \times 256$ for the Swin-TNP models, corresponding to a grid spacing of $\approx 1.41°$ in both the latitudinal and longitudinal directions. The results for the ConvCNP are for a grid size of $64 \times 128$, to maintain a smaller gap in parameter count between models.

**CNP**   We use a different deepset for the on- and the off-the-grid data, and the mean as the permutation-invariant function to aggregate the context tokens, i.e. $\mathbf{z}_c = \frac{1}{N_{\text{off},c}} \sum_{n=1}^{N_{\text{off},c}} e_{\text{off}}(\mathbf{x}_{\text{off},c,n}, \mathbf{y}_{\text{off},c,n}) + \frac{1}{N_{\text{on},c}} \sum_{n=1}^{N_{\text{on},c}} e_{\text{on}}(\mathbf{x}_{\text{on},c,n}, \mathbf{y}_{\text{on},c,n})$

**PT-TNP**   We managed to use up to $M = 256$ pseudo-tokens without running into memory issues. This shows that even if we only use two variables (one on- and one off-the-grid), PT-TNPs do not scale well to large data. We use a different encoder for the on- and off-the-grid data, before aggregating the two sets of tokens into a single context set.

**ConvCNP**   For the ConvCNP we use a grid of size $64 \times 128$ for all experiments. We first separately encode both the on- and the off-the-grid to the specified grid size using the SetConv. We then concatenate the two and project them to a dimension of $C = 128$ before passing through the U-Net (Ronneberger et al., 2015). The U-Net uses a kernel size of $k = 9$ with a stride of one.

**Swin-TNP**   For the Swin-TNP models in the main experiment, we use a grid size of $64 \times 128$, a window size of $4 \times 4$ and a shift size of $2 \times 2$. For the experiment with richer off-the-grid context sets (i.e. between $0.25$ and $0.5$ of the off-the-grid data), we use a grid size of $128 \times 256$, a window size of $8 \times 8$ and a shift size of $4 \times 4$.

### E.2.1 ADDITIONAL RESULTS FOR THE MAIN EXPERIMENT

We provide in Figure 18 a comparison for an example dataset between the predictive errors (i.e. difference between predicted mean and ground truth) produced by three models: Swin-TNP with PT-GE, ConvCNP, and PT-TNP. The predictions are performed at all station locations. The stations included in the context set are indicated with a black dot. The figures show how Swin-TNP usually produces lower errors in comparison to the baselines, indicated through paler colours. Examples of regions where this is most prominent include central US, as well as southern Australia and southern Europe.

**Analysis of the predictive uncertainties**   For the example dataset considered above, Figure 19 shows histograms of the normalised predictive errors, defined as the predictive errors divided by the predictive standard deviations. We compute the mean log-likelihoods under a standard normal distribution, and compare it to the reference negative entropy of the standard normal distribution of $-1.419$. This acts as an indicator of the accuracy of the predictive uncertainties outputted by the three models we consider: Swin-TNP (PT-GE, grid size $64 \times 128$), ConvCNP (grid size $64 \times 128$), and PT-TNP ($M = 256$). For the dataset considered in Figure 19, we obtain $-1.439$ for Swin-TNP, $-1.490$ for ConvCNP, and $-1.380$ for PT-TNP, indicating that, out of the three models, Swin-TNP outputs the most accurate uncertainties.

**Analysis of grid size influence**   We study to what extent increasing the grid size of the models, and hence their capacity, improves their predictive performance. We repeat the experiment for two

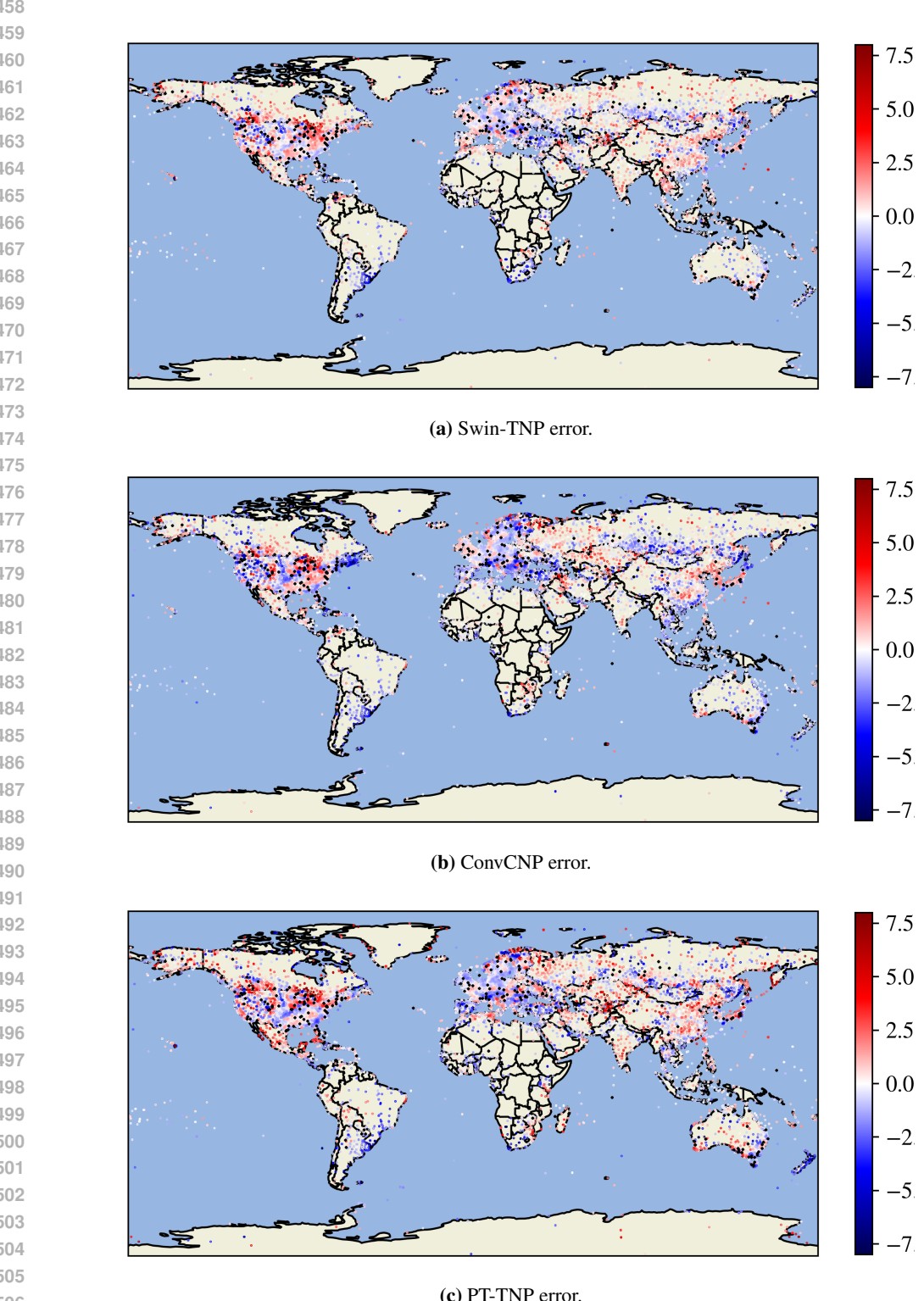

**(a)** Swin-TNP error.

**(b)** ConvCNP error.

**(c)** PT-TNP error.

**Figure 18:** A comparison between the predictive error—the difference between predictive mean and ground truth—of the 2m temperature at all weather station locations at 15:00, 28-01-2019. Stations included in the context dataset are shown as black dots ($3\%$ of all station locations). The mean predictive log-likelihoods (averaged across the globe) for these samples are $1.611$ (Swin-TNP, PT-GE, grid size of $64 \times 128$), $1.351$ (ConvCNP, grid size of $64 \times 128$), and $1.271$ (PT-TNP, $M = 256$).

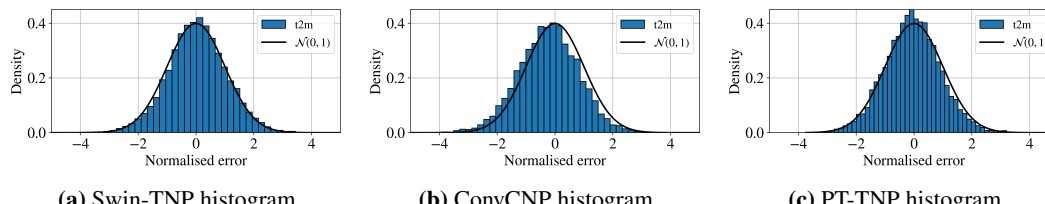

**(a)** Swin-TNP histogram.   **(b)** ConvCNP histogram.   **(c)** PT-TNP histogram.

**Figure 19:** A comparison between the normalised predictive error—the predictive error divided by the predicted standard deviation—of the 2m temperature at the US weather station locations at 15:00, 28-01-2019. The context set contains observations at $3\%$ of station locations. Each plot shows a histogram of the normalised errors based on the predictions at all station locations, alongside an overlaid standard normal distribution that perfect predictive uncertainties should follow. The mean log-likelihoods of the normalised predictive errors under a standard normal distribution for the Swin-TNP (PT-GE, grid size of $64 \times 128$), ConvCNP (grid size $64 \times 128$), and PT-TNP ($M = 256$) are $-1.439$, $-1.490$, and $-1.380$, respectively. For reference, a standard normal distributionn has a negative entropy of $-1.419$, indicating that the Swin-TNP has the most accurate predictive uncertainties.

models: Swin-TNP (with PT-GE) and ConvCNP with a grid size of $192 \times 384$, corresponding to $0.9375°$ in both latitudinal and longitudinal directions. For the Swin-TNP we use a window size of $8 \times 8$, and a shift size of $4 \times 4$. Due to time constraints, we only trained the Swin-TNP model for $280,000$ iterations instead of $300,000$. For the U-Net architecture within the ConvCNP we use a kernel size of nine. In the decoder of the bigger ConvCNP model, attention is performed over the nearest 9 neighbours, whereas for the smaller ConvCNP we use full attention. This makes the FPT of the bigger model smaller that that of the $64 \times 128$ model.

The results are shown in Table 5 (an represent a subset of the results shown in Table 1). In comparison to the Swin-TNP and ConvCNP models which use a grid size of $64 \times 128$, both models improve significantly. However, the bigger ConvCNP still underperforms both the small and big variant of Swin-TNP, with a significant gap in both log-likelihood and RMSE.

**Table 5:** Test log-likelihood ($\uparrow$) and RMSE ($\downarrow$) for the `t2m` station prediction experiment when varying grid size for two models. The standard errors of the log-likelihood are all below $0.004$, and of the RMSE below $0.005$. FPT: forward pass time for a batch size of eight in ms. Params: number of model parameters in units of M. Best results for each configuration (Swin-TNP / ConvCNP) are bolded.

| Model | GE | Grid size | Log-lik. $\uparrow$ | RMSE $\downarrow$ | FPT | Params |
|---|---|---|---|---|---|---|
| ConvCNP (no NN-CA) | SetConv | $64 \times 128$ | 1.535 | 1.252 | 96 | 9.36 |
| ConvCNP | SetConv | $192 \times 384$ | **1.689** | **1.166** | 74 | 9.36 |
| Swin-TNP | PT-GE | $64 \times 128$ | 1.819 | 1.006 | 127 | 2.29 |
| Swin-TNP | PT-GE | $192 \times 384$ | **2.053** | **0.873** | 306 | 10.67 |

**Analysis of influence of nearest-neighbour encoding and decoding**   A final ablation we perform in this experiment studies the influence of the number of nearest neighbours considered for the encoder and decoder on the performance of the model. Initially, we focus on the effect of nearest neighbour decoding, investigating two models—with and without full attention at decoding time. More specifically, we compare Swin-TNP with PT-GE and KI-GE with a grid size of $64 \times 128$, and either perform full attention in the grid decoder, or cross-attention over the 9 nearest neighbours (NN-CA). For the variants with full attention, we evaluate the log-likelihood at $25\%$ randomly sampled station locations instead of all of them because of memory constraints. The results are shown in Table 6 (but are also presented in Table 1), and indicate that, not only does NN-CA offer a more scalable decoder attention mechanism, but it also leads to improved predictive performance when applied to spatio-temporal data. We hypothesise this is due to the inductive biases it introduces, which are appropriate for the strong spatio-temporal correlations present in the data we used.

**Table 6:** Test log-likelihood (↑) and RMSE (↓) for the `t2m` station prediction experiment when varying the decoder attention mechanism—nearest-neighbour cross-attention and full attention (no NN-CA). The standard errors of the log-likelihood and RMSE are all below 0.003. FPT: forward pass time for a batch size of eight in ms. Params: number of model parameters in units of M. Best results for each configuration (PT-GE / KI-GE) are bolded.

| Model | GE | Grid size | Log-lik. ↑ | RMSE ↓ | FPT | Params |
|---|---|---|---|---|---|---|
| Swin-TNP | KI-GE | $64 \times 128$ | **1.683** | **1.157** | 121 | 1.14 |
| Swin-TNP (no NN-CA) | KI-GE | $64 \times 128$ | 1.544 | 1.436 | 137 | 1.14 |
| Swin-TNP | PT-GE | $64 \times 128$ | **1.819** | **1.006** | 127 | 2.29 |
| Swin-TNP (no NN-CA) | PT-GE | $64 \times 128$ | 1.636 | 1.273 | 144 | 2.29 |

Focusing on just the models using the PT-GE, we also study intermediate regimes for the nearest neighbour decoding mechanism with $k_{\text{dec}} = 25$, and $k_{\text{dec}} = 49$. Moreover, we also investigate how the models perform with an increased number of nearest neighbours considered during encoding ($k_{\text{enc}} = 9$). The full results are presented in Table 7, where for each model we specify the number of nearest neighbours considered in the encoder ($k_{\text{enc}}$) and decoder ($k_{\text{dec}}$). The training for these models is still on-going, but we present the number of training iterations (out of 300k) in Table 7, and we will update the figures once training is finished.

**Table 7:** Test log-likelihood (↑) and RMSE (↓) for the `t2m` station prediction experiment when varying the number of nearest neighbours considered for the encoder ($k_{\text{enc}}$) and decoder ($k_{\text{dec}}$). No NN-CA signifies that full attention is applied in the decoder. For the models that have not finished training, we indicate between brackets the number of training iterations. The standard errors of the log-likelihood and RMSE are all below 0.003. FPT: forward pass time for a batch size of eight in ms. Params: number of model parameters in units of M. Best results for each configuration (PT-GE / KI-GE) are bolded.

| Model | GE | $k_{\text{enc}}$ | $k_{\text{dec}}$ | Grid size | Log-lik. ↑ | RMSE ↓ | FPT | Params |
|---|---|---|---|---|---|---|---|---|
| Swin-TNP | PT-GE | 1 | 9 | $64 \times 128$ | **1.819** | **1.006** | 127 | 2.29 |
| Swin-TNP (no NN-CA) | PT-GE | 1 | - | $64 \times 128$ | 1.636 | 1.273 | 144 | 2.29 |
| Swin-TNP (260k) | PT-GE | 1 | 25 | $64 \times 128$ | 1.778 | 1.050 | 211 | 2.29 |
| Swin-TNP (215k) | PT-GE | 1 | 49 | $64 \times 128$ | 1.714 | 1.097 | 306 | 2.29 |
| Swin-TNP (230k) | PT-GE | 9 | 9 | $64 \times 128$ | **1.843** | **1.002** | 360 | 2.29 |

We observe that the performance of the models tends to:

- Slightly degrade with increasing $k_{\text{dec}}$—we believe this is because locality represents a good inductive bias in the task we consider. We also suspect that with sufficient training the models with different $k_{\text{dec}}$ would eventually reach similar performance, but a lower value encourages more efficient training.
- Slightly improve with increasing $k_{\text{enc}}$—the gridded pseudo-tokens are, on average, modulated by more context points, hence increasing predictive performance. However, this comes at an increased computational cost.

### E.2.2 ADDITIONAL RESULTS FOR RICHER CONTEXT SET

In order to investigate whether the model manages to learn meaningful relationships between the off- and on-the-grid data (`t2m` and `skt`) and to exploit the on-the-grid information[13], we also perform an experiment with richer context sets. More specifically, the number of off-the-grid context points is sampled according to $N_{\text{off},c} \sim \mathcal{U}_{[0.25,0.5]}$. The results are given in Table 8.

For the ConvCNP we evaluated two models—one with full decoder attention and one with nearest-neighbour cross-attention (NN-CA). Similarly to the previous section, we found that the latter has a

---

[13]This is achieved by comparing the performance of a model that is only given off-the-grid context information, with a similar model that is provided with both off- as well as on-the-grid data.

better performance with a log-likelihood of $1.705$ (NN-CA) as opposed to $1.635$ (full attention). As such, Table 8 shows the results for the NN-CA ConvCNP model.

**Table 8:** Test log-likelihood ($\uparrow$) and RMSE ($\downarrow$) for the t2m station prediction experiment with richer off-the-grid context information. The standard errors of both the log-likelihood and the RMSE are all below $0.003$. FPT: forward pass time for a batch size of eight in ms. Params: number of model parameters in units of M.

| Model | GE | Grid size | Log-lik. ($\uparrow$) | RMSE ($\downarrow$) | FPT | Params |
|---|---|---|---|---|---|---|
| CNP | - | - | 0.715 | 3.056 | 27 | 0.34 |
| PT-TNP | - | $M = 256$ | 1.593 | 1.403 | 219 | 1.57 |
| ConvCNP | SetConv | $64 \times 128$ | 1.705 | 1.100 | 21 | 9.36 |
| ViTNP | KI-GE | $48 \times 96$ | 1.754 | 1.112 | 175 | 1.14 |
| ViTNP | PT-GE | $48 \times 96$ | 1.988 | 0.932 | 188 | 1.83 |
| ViTNP | KI-GE | $144 \times 288 \to 48 \times 96$ | 1.842 | 1.046 | 179 | 1.29 |
| ViTNP | PT-GE | $144 \times 288 \to 48 \times 96$ | 2.242 | 0.798 | 221 | 6.69 |
| Swin-TNP | KI-GE | $128 \times 256$ | 2.362 | 0.758 | 174 | 1.14 |
| Swin-TNP | PT-GE | $128 \times 256$ | **2.446** | **0.697** | 208 | 5.43 |
| Swin-TNP (skt) | PT-GE | $128 \times 256$ | 1.501 | 1.266 | 178 | 5.43 |
| Swin-TNP (t2m) | PT-GE | $128 \times 256$ | 2.331 | 0.909 | 186 | 5.38 |

The results are consistent with the findings from the main experiment:

- The performances of the baselines (CNP, PT-TNP, and ConvCNP) are significantly worse than the gridded TNP variants considered.

- For each gridded TNP variant, the pseudo-token grid encoder (PT-GE) performs better than the kernel-interpolation one (KI-GE).

- The variants with a Swin-transformer backbone outperform the ones with a ViT-backbone, even when the latter has more parameters and a higher FPT.

- Among the ViT variants, the ones that employ patch encoding before projecting to a $48 \times 96$ grid outperform the ones that directly encode to a $48 \times 96$ grid.

- Performing nearest-neighbour cross-attention in the decoder as opposed to full attention leads to both computational speed-ups, as well as enhanced predictive performance.

In comparison to the main experiment, the gap in performance between Swin-TNP and Swin-TNP (t2m) is smaller—this is expected, given that the context already includes between $25\%$ and $50\%$ of the off-the-grid station locations. However, the gap is still significant, implying that Swin-TNP manages to leverage the on-the-grid data (skt) and exploits its relationship with the target t2m to improve its predictive performance.

### E.3 COMBINING MULTIPLE SOURCES OF UNSTRUCTURED WIND SPEED OBSERVATIONS

In this experiment, we consider modelling the eastward (u) and northward (v) components of wind speed at 700hPa, 850hPa and 1000hPa (surface level). These quantities are essential for understanding and simulating large-scale circulation in the atmosphere, for wind energy integration into power plants, or for private citizens and public administrations for safety planning in the case of hazardous situations (Lagomarsino-Oneto et al., 2023). We obtain each of the six modalities from the ERA5 reanalysis dataset (Hersbach et al., 2020), and construct datasets over a latitude / longitude range of $[25°, 49°] / [-125°, -66°]$, which corresponds to the contiguous US, spanning four hours. We show plots of wind speeds at the three pressure levels for a single time step in Figure 20.

For each task, we first sample a series of four consecutive time points. From this $4 \times 96 \times 236$ grid, we sample a proportion $p_c$ and $p_t$ of total points to form the context and target datasets, where $p_c \sim \mathcal{U}_{[0.05, 0.25]}$ and $p_t = 0.25$. All models are trained for $300,000$ iterations on hourly data between $2009 - 2017$ with a batch size of eight. Validation is performed on 2018 and testing on 2019. Experiment specific architecture choices are described below.

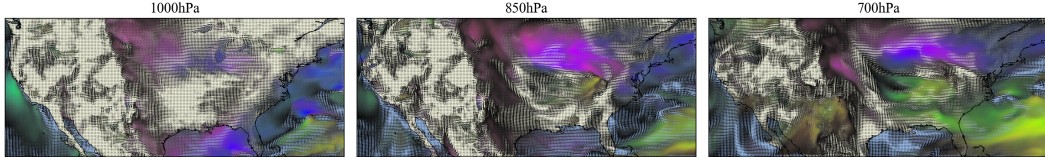

**Figure 20:** Wind-speed and direction at each of the three pressure levels, 700hPa, 850hPa and 1000hPa, over the contiguous US at 15:00 GMT, 08-06-1997. The colours correspond to the magnitude and direction of the wind speed.

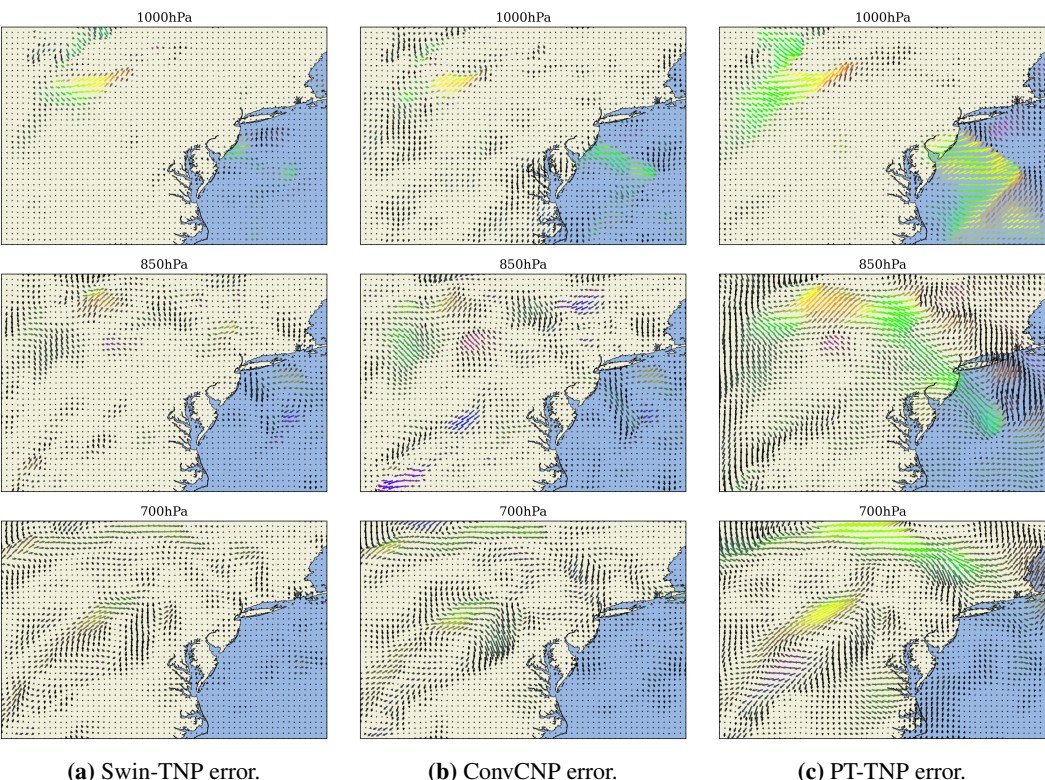

      **(a)** Swin-TNP error.           **(b)** ConvCNP error.           **(c)** PT-TNP error.

**Figure 21:** A comparison between the predictive error—the difference between predictive mean and ground truth—of normalised wind speeds for a selection of CNP models on a small region of the US at 04:00, 01-01-2019. Each plot consists of 2,400 arrows with length and orientation corresponding to the direction and magnitude of the wind-speed error at the corresponding pressure level. The colour of each arrow is given by the HSV values with hue dictated by orientation, and saturation and value dictated by length (i.e. the brighter the colour, the larger the error). For this dataset, the context dataset consists of $5\%$ of the total available observations, and the corresponding mean predictive log-likelihoods for the Swin-TNP (PT-GE, grid size $4 \times 24 \times 60$), ConvCNP (grid size $4 \times 24 \times 60$) and PT-TNP ($M = 64$) are $5.84$, $3.23$ and $2.41$.

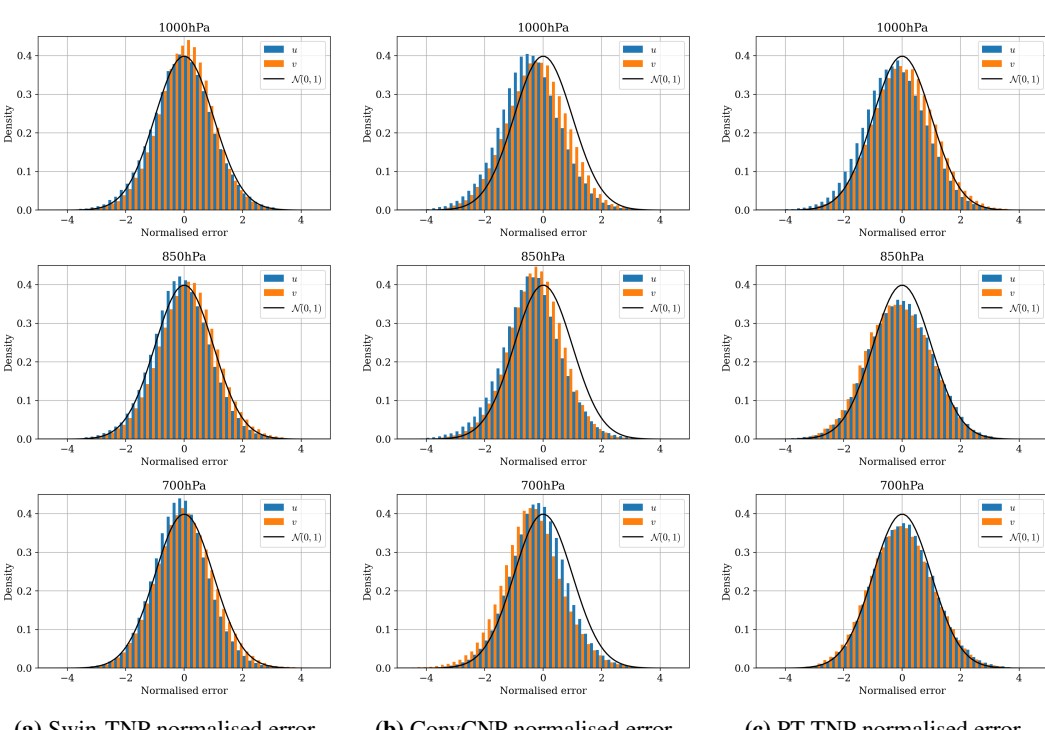

**(a)** Swin-TNP normalised error.  **(b)** ConvCNP normalised error.  **(c)** PT-TNP normalised error.

**Figure 22:** A comparison between the normalised predictive error—the predictive error divided by the predictive standard deviation—of wind speeds for a selection of CNP models on a small region of the US at 04:00, 01-01-2019. Each plot compares a histogram of values for both the u and v components with a standard normal distribution, which perfect predictive uncertainties follow. For this dataset, the context dataset consists of $5\%$ of the total available observations. The mean log-likelihoods (averaged over pressure levels and the 4 time points) of the normalised predictive errors under a standard normal distribution for the Swin-TNP (PT-GE, grid size $4 \times 24 \times 60$), ConvCNP (grid size $4 \times 24 \times 60$) and PT-TNP ($M = 64$) are -1.425, -1.501 and -1.514. For reference, a standard normal distribution has a negative entropy of -1.419. This indicates that the Swin-TNP has the most accurate predictive uncertainties.

**Input embedding**    As the input contains both temporal and latitude / longitude information, we use both Fourier embeddings for time and spherical harmonic embeddings for the latitude / longitude values. These are not used in the ConvCNP as the ConvCNP does not modify the inputs to maintain translation equivariance (in this case, with respect to time and the great-circle distance).

**Grid sizes**    We chose a grid size of $4 \times 24 \times 60$ for the ConvCNP and Swin-TNP models, as this corresponds to a grid spacing of $1°$ in the latitudinal direction and around $1°$ in the longitudinal direction. For the ViTNP, we chose a grid size of $4 \times 12 \times 30$, corresponding to a grid spacing of $2°$.

**CNP**    A different deepset is used for each modality, with the aggregated context token for each modality then summed together to form a single aggregated context token, i.e. $\mathbf{z}_c = \sum_{s=1}^{S} \frac{1}{N_{c,s}} \sum_{n=1}^{N_{c,s}} e_s(\mathbf{x}_{c,n,s}, \mathbf{y}_{c,n,s})$.

**PT-TNP**    For this experiment, we could only use $M = 64$ pseudo-tokens for the PT-TNP without running into out-of-memory issues. This highlights a limitation in scaling PT-TNPs to large datasets. We note that there does exist work that remedies the poor memory scaling of PT-TNPs (Feng et al., 2024); however, this trades off against time complexity which itself is a bottleneck given the size of datasets we consider. A different encoder is used for each modality, before aggregating the tokens into a single context set of tokens.

**ConvCNP**    For the ConvCNP, we use a grid size of $4 \times 24 \times 60$. Each modality is first grid encoded separately using the SetConv, concatenated together and then to $C = 128$ dimensions before passing through the U-Net.

**Swin-TNP**    For the Swin-TNP models, we use a grid size of $4 \times 24 \times 60$, a window size of $4 \times 4 \times 4$ and a shift size of $0 \times 2 \times 2$.

