# OpenReview forum: "Gridded Transformer Neural Processes for Large Unstructured Spatio-Temporal Data"
_ICLR.cc/2025/Conference — Submitted to ICLR 2025_

### Official Review · Reviewer_VoCT · 2024-11-03

**Soundness:** 3
**Presentation:** 3
**Contribution:** 3
**Rating:** 6
**Confidence:** 4

**Summary:**

Transformers have shown promising results in weather forecasting but are limited to gridded structured data and neglect large amounts of unstructured observational data.
Transformer Neural Processes (TNPs) are more flexible, capable of handling both structured and unstructured data, but they struggle with scalability.
To address this issue, this paper introduces gridded pseudo-token TNPs.
Gridded TNPs consists of an encoder to convert the unstructured observation data to a structured grid via cross attention, a transformer (i.e., Vision Transformer [ViT] or Swin Transformer)-based processor to generate gridded token prediction, and a decoder with k-nearest-neighbor cross attention to map the predicted token back to the unstructured data at arbitrary locations.
The paper evaluates the methods on a few benchmarks, comparing the pseudo-token grid encoder to the kernel interpolation approaches and ViT to Swin Transformer.

**Strengths:**

The proposed gridded pseudo-token encoder and decoder with k-nearest-neighbors via cross-attention provided a flexible and elegant way to deal with both unstructured and structured data. The paper is well written, and the method is clearly presented.

**Weaknesses:**

1. The experiment results are not analyzed in detail enough to substantiate the claims. The reviewer had a difficult time to link the very general discussions in section 5.4 to the experiments in sections 5.1-5.3. Adding a few sentences on the observations/conclusions of each experiment would be helpful.
2. The claim that introducing ViT and Swin Transformer into the transformer backbone as an efficient mechanism is a main contribution feels somewhat weak.
2. Typos:
	1) In Fig. 2, the “{(x_{c,n},z_{c,n})}_n=1^N_c” should be “{(z_{c,n})}_n=1^N_c”?
	2) Line 344 on page 7, “e_s(x_{c,n,s},x_{c,n,s})” should be “e_s(x_{c,n,s},y_{c,n,s})”?

**Questions:**

1. GNNs are another type of commonly used approaches for handling unstructured data? Could the authors comment on them?
2. Compared to KI-GE, PT-GE appears to achieve better accuracy but requires a higher FPT. In the ViTNP cases in Table 1, the FPTs are 215 vs. 171 for PT-GE and KI-GE, respectively. Could the authors provide an explanation for this difference in time cost?

---

> ### Author Response · Authors · 2024-11-18
> **Response to Reviewer VoCT**
>
> Many thanks for your review and for the concerns you have raised, in particular with regards to providing additional clarity into how we arrived at our conclusions given the results. We address each of them below.
>
> ***“The experiment results are not analyzed in detail enough to substantiate the claims.”***
>
> Thank you for this valuable feedback. While our original approach of unified discussion aimed for conciseness—given that the takeaways across experiments are consistent—we agree that this structure could be more reader-friendly.
>
> In the revised version, we:
> 1. Added a focused discussion after each experiment that explicitly connects those specific results to our claims.
> 2. Maintain the unified discussion section, but introduce it as a summary of the findings across the three experiments.
>
> ***“The claim that introducing ViT and Swin Transformer into the transformer backbone… feels somewhat weak.”***
>
> We agree that our text could better articulate our key contributions. To clarify: our innovation is not the mere use of ViT and Swin Transformer architectures, but rather developing a novel framework that *enables* these powerful grid-based architectures to process unstructured data. We revised the manuscript to emphasise this distinction.
>
> ***“{(x_{c, n}, z_{c, n})}n=1^N_c should be {(z{c,n})}_n=1^N_c”***
>
> Our notation ${(x_{c, n}, z_{c, n})}$ is *intentional*. The spatial locations $x_{c, n}$ of the context tokens are essential to our model’s operation in two key ways:
> 1. They are used in the Swin Transformer to construct appropriate window partitions.
> 2. They are required in the grid decoder for spatial interpolation.
>
> We will add a clarifying note to the caption of Figure 2 to make this dual purpose explicit.
>
> ***“e_s(x_{c,n,c},x_{c,n,x}) should be e_s(x_{c,n,s},y_{c,n,s})”***
>
> Great spot, thank you! We have corrected this typo.
>
> ***“GNNs… Could the authors comment on them”***
>
> We are pleased that you brought up the connections between transformer-based architectures and GNNs, specifically for handling unstructured data. The two different approaches are intimately related, and we will do our best to provide an overview of this relationship, and the limitations thereof.
>
> 1. **Computational complexity challenges.**
>     * Fully-connected GNNs, like regular transformers, face quadratic complexity with respect to the number of nodes (number of observations)
>     * While sparse connectivity (used in GraphCast and GenCast) helps with gridded data, it poses challenges for unstructured data. In particular, to induce sparse connectivity we would require dynamically computing $k$ nearest neighbours for each node, which a straightforward implementation of demands quadratic computational complexity.
> 2. **Architectural equivalence.**
> Our approach can be mapped to a GNN equivalent as follows:
>     * The pseudo-token grid encoder = GNN connecting observations to the nearest nodes on some latent grid of pseudo-tokens with directed edges.
>     * The Swin Transformer processing = GNN with fixed sparse connectivity between nodes within each window and staggered GNN updates.
>     * The grid decoder  = GNN connecting a latent grid of pseudo-tokens to target prediction locations with directed edges.
> 3. **Practical considerations.**
>     * While theoretically very similar, empirical evidence suggests that transformer-based implementations achieve superior computational efficiency compared to their GNN counterparts (e.g. see discussion in second paragraph of Section 2 of [1]).
>     * Furthermore, the widespread adoption of transformer architectures has led to transformer-specific operations, such as attention mechanisms, being highly optimised for modern GPU architectures [2] and seamlessly integrable, making transformers an efficient and convenient choice from a practical perspective.
>
> If you think it will add value to the paper, we are happy to add a discussion of these connections in the Appendix.
>
> [1] Lang et al. AIFS - ECMWF’s Data-Driven Forecasting System. arXiv preprint arXiv:2406.01465 (2024).
>
> [2] Dao, T. FlashAttention-2: Faster Attention with Better Parallelism and Work Partitioning. arXiv preprint arXiv:2307.08691 (2023).
>
> ***“Could the authors provide an explanation for this difference in time cost”***
>
> The difference in computational cost can be attributed to the cross-attention operation being more computationally expensive than a kernel-interpolation operation. This is due to cross-attention operations requiring additional overhead costs, such as those associated with passing tokens pointwise through an MLP within the cross-attention layer.

---

> ### Author Response · Authors · 2024-11-21
> **Reminder of our Rebuttals**
>
> Dear Reviewer VoCT,
>
> We appreciate that you are likely to be reviewing multiple other papers; however, as we are approaching the end of the discussion period (less than one week remaining), we would greatly appreciate your feedback on our rebuttals. Your additional insights would be valuable in helping us improve the paper further.
>
> Kind regards,
> The authors of paper 9931

---

### Official Review · Reviewer_zSqu · 2024-11-03

**Soundness:** 3
**Presentation:** 3
**Contribution:** 3
**Rating:** 8
**Confidence:** 4

**Summary:**

The paper proposes Gridded Transformer Neural Processes (Gridded TNPs), an extension of TNPs that scales better to large-scale datasets. The idea is to use pseudo tokens and cross-attention to bring unstructured, point-based data to a grid before further processing by efficient on-grid architectures such as ViTs and Swin Transformers, and finally decoding to off-grid predictions via cross-attention. Different variants of Gridded TNPs were also introduced with different grid encoding methods and processor architectures. Gridded TNPs outperform the baselines on synthetic regression and station-based weather forecasting.

**Strengths:**

Overall, I think the ability to blend off-grid and on-grid data of Gridded TNPs contributes well to the spatiotemporal modeling community. I like the simplicity of the proposed method and the clear writing of the paper. The experiments, even though are still at small scales, are reasonable for an academic setting, and sufficiently show the good performance of Gridded TNPs.

**Weaknesses:**

I do not see any critical weakness in the paper at the moment, but would like some clarifications and more analyses of the design choices in the paper. Specifically:
- What are the queries, keys, and values in the cross-attention operation in the grid encoder? What I understand from the paper is you use the learnable pseudo tokens as queries and embeddings of context points as keys and values. If this is true, do you use any positional encoding for the pseudo tokens?
- The paper said k=1 for the grid encoder is sufficient. Does this mean each pseudo token only attends to one context point that is closest to its grid location? What is the difference between this and simply picking the closest context point?
- In the grid decoder, where do the target tokens Z_t come from? Are they embeddings of the target inputs x's?
- I'd like to see more ablation studies about choosing k in both the encoder and decoder across different datasets and tasks.

----- POST REBUTTAL -----

The authors have addressed my main concerns so I'm raising my score to 8.

**Questions:**

See above.

---

> ### Author Response · Authors · 2024-11-18
> **Response to Reviewer zSqu**
>
> We thank the reviewer for their comments, and address each of the concerns below. As we note, many of the concerns and questions are already addressed in the paper. We therefore hope the reviewer might consider increasing their score.
>
> ***“What are the queries, keys, and values in the cross-attention operation in the grid encoder”***
>
> Thank you for your clarifying question. The cross-attention operation in our grid encoder is precisely defined in Equations 3 and 4, where:
> 1. *Queries*: the pseudo-tokens, each representing the token value at a fixed pseudo-location on a grid.
> 2. *Keys*: The set of context tokens for which the pseudo-token is amongst the nearest $k$ neighbours.
> 3. *Values*: The same set of context tokens used as keys.
>
> Importantly, we do not need explicit positional encodings for the pseudo-tokens because:
> 1. Each pseudo-token corresponds to a fixed spatio-temporal location in the grid.
> 2. The pseudo-tokens are treated as learnable parameters.
> 3. Through optimisation, the model implicitly learns the appropriate spatial-temporal relationships, effectively encoding positional information into the pseudo-token values themselves.
> 4. This design choice simplifies the architecture while maintaining the model’s ability to capture spatio-temporal relationships in the data.
>
> ***“Does this mean each pseudo-token only attends to one context point that is closest to its grid location?”***
>
> The parameter $k=1$ (discussed on line 289) actually implies that each pseudo-token attends to the set of context tokens for which *it* is the closest pseudo-token, not the other way around. To illustrate with a simple 1-D example, suppose that ‘$o(n)$’ denotes the $n$-th context token and ‘$x(m)$’ denotes the $m$-th pseudo-token, and we have the following structure of context and pseudo-tokens:
>
> $$o(1) o(2) x(1) x(2) o(3) o(4) x(3)$$
>
> Then, the attention relationships are as follows:
> 1. Pseudo-token $x(1)$ will attend to $(o(1), o(2))$ (as it’s the nearest pseudo-token to both).
> 2. Pseudo-token $x(2)$ attends to $(o(3))$.
> 3. Pseudo-token $x(3)$ attends to $(o(4))$.
>
> This mechanism is visualised in Figure 3 of our paper. It ensures that each pseudo-token integrates information from all relevant context tokens in its neighbourhood, not just a single closest point.
>
> ***“...where do the target tokens Z_t come from?”***
>
> You are correct: they are embeddings of the target inputs $x_t$. We discuss this in lines 170-171 in Section 2.2.3.
>
> ***“...more ablation studies about choosing k in both the encoder and decoder…”***
>
> We appreciate this suggestion, and agree. We have performed a set of additional experiments for the weather station experiment which explores this in greater depth. These are shown in Appendix E.2.1 - Analysis of influence of nearest-neighbour encoding and decoding (highlighted in red), alongside a short discussion about the results.
>
> While we agree that it is interesting to perform these ablations across different datasets and tasks, these experiments are expensive to run and we only have limited time available to provide the results. This is the reason we did not run them on the multi-modal experiment too. However, the influence of these parameters depending on the task we are considering is something we are willing to explore further.

---

> ### Author Response · Authors · 2024-11-21
> **Reminder of our Rebuttals**
>
> Dear Reviewer zSqu,
>
> We appreciate that you are likely to be reviewing multiple other papers; however, as we are approaching the end of the discussion period (less than one week remaining), we would greatly appreciate your feedback on our rebuttals. Your additional insights would be valuable in helping us improve the paper further.
>
> Kind regards,
> The authors of paper 9931

---

> > ### Comment · Reviewer_zSqu · 2024-11-22
> >
> > Thank you for the clarifications. They have addressed my main concerns. I have updated my score to 8.

---

> > > ### Author Response · Authors · 2024-11-22
> > >
> > > Many thanks for responding to our rebuttal and raising your score in turn. We greatly appreciate your feedback on our paper---thank you.

---

### Official Review · Reviewer_7UA7 · 2024-11-04

**Soundness:** 3
**Presentation:** 3
**Contribution:** 3
**Rating:** 5
**Confidence:** 3

**Summary:**

This paper introduces a Gridded Transformer Neural Process (G-TNP), which effectively maps unstructured spatiotemporal data onto a grid and incorporates an efficient attention mechanism, enhancing predictive performance on large-scale datasets.

**Strengths:**

The paper systematically introduces the G-TNPs model architecture, including the grid encoder, grid processor, and grid decoder, each accompanied by detailed mathematical descriptions and theoretical foundations.

**Weaknesses:**

1. I'm curious, have you considered using large-scale meteorological models like Pangu as a baseline for comparison?
2. Although an efficient attention mechanism has been introduced, the model's complexity might still be high, especially when handling ultra-large-scale datasets. Could the authors provide comprehensive data to clarify this?

**Questions:**

See Weaknesses.

---

> ### Author Response · Authors · 2024-11-18
> **Response to Reviewer 7UA7**
>
> Many thanks for your review. We appreciate your questions, and we answer both below. If there is anything else we can clarify then please let us know. If not, and you feel that the contributions of our work are strong enough, then we encourage you to please raise your score to recommend acceptance.
>
> ***“...have you considered using large-scale meteorological models like Pangu…”***
>
> As we discuss throughout the paper, a primary motivation for this research is that the current fleet of large-scale meteorological models like Pangu are fundamentally limited to gridded data structures. This constraint prevents us from evaluating these models on unstructured data, which is what we use in our experiments, and so we cannot use them as baselines.
>
> This limitation is particularly significant for weather and climate modelling, where raw observations rarely conform to a regular grid. Our work addresses this challenge in two key ways:
> 1. We develop novel methods to transform unstructured data into a structured grid of pseudo-tokens, making them compatible with established architectures like ViT and Swin Transformer.
> 2. We introduce techniques to evaluate predictions at arbitrary target locations, not just the gridded locations.
>
> These innovations enable our approach to bridge the gap between irregularly sampled real-world observations and grid-based deep learning methods. .
>
> ***“...the model’s complexity might still be high…”***
>
> We appreciate the concern about model complexity. While our model’s computational requirements do scale with dataset size, this scaling is both predictable and manageable. Specifically:
> 1. The computational complexity is dominated by our grid processing components (ViT/Swin Transformer)---the same architectures successfully deployed in production-scale meteorological models.
> 2. Crucially, our model’s complexity scales *linearly* with dataset size: see lines 295-296 and line 333 for the computational complexity of the grid encoder and grid decoder, and the computational complexity of the grid processor is dependent on the number of pseudo-tokens, which can be freely chosen but generally scales linearly with the dataset size. This allows for precise resource planning when scaling to larger datasets.
> 3. These architectures have demonstrated robust performance on ultra-large datasets when provided with appropriate computational resources (e.g., Aurora [1], AIFS [2], Pangu [3], ClimaX [4]).
> 4. This linear scaling relationship is particularly advantageous compared to potential alternatives that might exhibit quadratic or higher-order scaling behaviours.
>
> [1] Bodnar et al. Aurora: A Foundation Model of the Atmosphere. arXiv preprint arXiv:2405.13063 (2024).
>
> [2] Lang et al. AIFS - ECMWF’s Data-Driven Forecasting System. arXiv preprint arXiv:2406.01465 (2024).
>
> [3] Bi et al. Accurate medium-range global weather forecasting with 3D neural networks. Nature (2023).
>
> [4] Nguyen et al. ClimaX: A foundation model for weather and climate. ICML (2023).

---

> ### Author Response · Authors · 2024-11-21
> **Reminder of our Rebuttals**
>
> Dear Reviewer 7UA7,
>
> We appreciate that you are likely to be reviewing multiple other papers; however, as we are approaching the end of the discussion period (less than one week remaining), we would greatly appreciate your feedback on our rebuttals. Your additional insights would be valuable in helping us improve the paper further.
>
> Kind regards,
> The authors of paper 9931

---

### Official Review · Reviewer_XaBC · 2024-11-05

**Soundness:** 1
**Presentation:** 3
**Contribution:** 2
**Rating:** 3
**Confidence:** 4

**Summary:**

A gridded transformer architecture is introduced that builds on neural processes. The model builds on three stages. First, unstructured data is encoded into a grid, second the grid is processed with an efficient ViT or SwinTransformer architecture, and third the grid is decoded to provide predictions at arbitrary positions beyond the grid. A comparison to other deep learning methods for processing unstructured grids demonstrates the benefits of the introduced method.

**Strengths:**

_Originality:_ The presented approach is original by combining established methods, such as Transformer Neural Processes, with efficient attention mechanisms from ViT and SwinTransformer, to tackle a highly relevant real-world problem. It would be very interesting to see, how the proposed method compares to ERA5 data and to understand the biases this approach imposes. Coming up with a data driven ERA5 alternative that mitigates numerical model biases would be of extraordinary value.

_Quality:_ The consideration of uncertainties in the output is a great design choice for the task of working with atmospheric states. Figures are mostly well designed and support the messages conveyed in the manuscript. Honestly, I do not see the value of Figure 1, in particular, since Figure 2 outlines the same pipeline both more detailed and concrete.

_Clarity:_ The manuscript is well organized, structured, and written. A rich appendix provides details about the model design, and a supplementary material with code is provided.

**Weaknesses:**

Overall, the paper touches on many interesting aspects, but misses a central message and problem that is tackled. Advertising the model more towards an alternative of traditional data assimilation techniques via numerical weather prediction models could be an interesting focus.

_Significance:_ It is hard to assess, whether the introduced method offers advantages over traditional data assimilation methods. To this end, the manuscript reads like introducing a neural network for data driven data aggregation (i.e., an alternative to ERA5), which potentially would provide substantial value. I would therefore suggest a focus shift from minimizing forecast error, such as done by all these deep learning weather prediction models, to generating a coherent state of the atmosphere, which is much more novel. Comparing to ERA5 would be highly appreciated, even though one cannot expect to outperform this dataset yet. Reporting first steps would be very insightful, though, and of large interest for the community.

_Quality:_
- **Qual1** Apart from the positive quality aspects mentioned under "Strengths," the manuscript can be improved in terms of supporting claims. That is, neural processes are introduced as flexible to downstream tasks, such as forecasting, data fusion, data interpolation and data assimilation (ll. 60--61). However, the model is only evaluated in terms of diagnosis error but not on the quality of other tasks.
- **Qual2** The use of different encoders for each input modality (l. 342) sounds inspired by ClimaX, which would be fair to be referenced here.
- **Qual3** Due to scaling reasons, the introduced method is not compared to its predecessors ANP and TNP (l. 366). However, it would be valuable to understand how the introduced method performs on a small dataset, where ANP and TNP are applicable, to understand whether the decign choices made in the new model result in performance decrease, and if so, to what extent.
- **Qual4** The manuscript advertises the transition from gridded reanalysis dataset, such as ERA5, from numerical weather prediction models. Yet, the model depends on ERA5 as additional gridded input beyond the heterogeneously distributed weather station data. It is unclear, how the gridded data and the weather station data is integrated and why the gridded data is necressary.
- **Qual5** GraphCast and GenCast follow a similar processing pipeline, where data points are aggregated into a grid, then processes, and subsequently decoded. A comparison to the aggregation approaches in these methods would be insightful. Otherwise it is hard to assess the quality of the introduced method over well established approaches.
- **Qual6** Figure 14 is very hard to interpret. A colourmap as well as a difference plot to the ground truth would help substantially to assess the quality of the different models. Similarly, it is difficult to extract the content of Figure 16, which is too dense. Zooming into certain regions could help.

_Clarity:_ Readers might benefit if the prediction task in Section 5.2 and Section 5.3 was introduced as diagnosis, i.e., diagnosing from some input stations in time $t$ to other input stations in the same time step $t$, where no forecasting is involved.

**Questions:**

1. How is the latent grid structured? That is, what resolution is chosen and how are the grid points distributed over the sphere, e.g., equirectangularly? Have you considered distortions towards polar regions in ERA5 data? I could find some information in Table 4 of the Appendix, but could not see for which tasks these grids were chosen (i.e., global vs local diagnosis).
2. What effect does the size of the attention circle (or diameter) in the decoder have on the performance? Depending on the target variable, this might be a crucial hyperparameter. However, increasing the attention radius increases the computation time. Can you provide some information on the relevance of the size of the attention circle for different target variables (e.g., t2m vs. wind) and what computational cost the size of the decoder's attention circle imposes?

---

> ### Author Response · Authors · 2024-11-18
> **Response to Reviewer XaBC (part 1)**
>
> We sincerely appreciate your thorough review and detailed feedback. However, we believe some aspects of our paper’s scope and contributions have been misunderstood. Below, we delineate both what our paper does and does not aim to accomplish.
>
> **What This Paper Is**
>
> 1. **Enables Transformers to Efficiently handle Unstructured Spatio-Temporal Data**
>     * Novel method to apply highly efficient ViT and  Swin Transformers to non-gridded spatio-temporal data.
>     * Allows efficient transformer architectures to be applied to irregularly sampled data.
> 2. **An Extension of TNPs**
>     * Addresses quadratic complexity limitations associated with regular TNPs.
>     * Enables efficient attention mechanisms to be used.
>     * Improves scalability.
> 3. **A General-Purpose Framework**
>     * Parameterises dataset-to-distribution mappings. That is, models that map directly from datasets—a collection of input-output pairs in the case of supervised learning—to predictive distributions at arbitrary input locations.
>     * Applicable to multiple tasks (e.g. data assimilation, regression, classification).
>     * Provides probabilistic predictions crucial for many tasks.
> 4. **A New Method which is Evaluated Using ERA5**
>     * ERA5 is chosen for convenience and accessibility. As mentioned, it is widely used for evaluating machine learning models that are similar to ours, e.g. [1]-[7].
>     * Used as ground-truth for evaluation, not as a predictive comparison.
>     * Demonstrates practical applicability and scalability of our method.
>
> **What This Paper Is Not**
>
> 1. **The Paper Does Not Aim to Provide an Alternative to ERA5**
>     * Our goal is not to replace ERA5, but rather we utilise it as a datasource for evaluating our model.
>     * ERA5 is widely used for evaluating machine learning methods that are similar to ours, e.g. [1]-[7].
> 2. **The Paper Is Not Weather-Specific**
>     * Our model is *domain-agnostic*.
>     * Weather experiments serve as validation, not the only application of our methods.
>     * Custom architectures would be needed for specific domain applications.
> 3. **The Paper Is Not a Competitor to GraphCast/GenCast**
>     * These models address *structured data forecasting*, where data lies on a regular grid or mesh.
>     * Our work extends transformer-based models in general (specifically TNPs).
>     * We focus on enabling efficient-attention-based architectures, such as ViT and Swin Transformer, for *unstructured data*.
> 4. **The Paper Is Not Focused on Forecasting**
>     * We address general prediction tasks: given a dataset of observations at some locations, predict the observations at a different set of locations.
>     * One can formulate specific problems by choosing which locations and quantities to evaluate.
>     * Our scope includes prediction at any spatio-temporal location.
>     * Forecasting is just one possible application.
>
> **References**
>
> [1] Gordon et al. Convolutional Conditional Neural Processes. ICLR (2020).
>
> [2] Foong et al. Meta-Learning Stationary Stochastic Process Prediction with Convolutional Neural Processes. NeurIPS (2020).
>
> [3] Scholz et al. Sim2Real for Environmental Neural Processes. arXiv preprint arXiv:2310.19932 (2023)
>
> [4] Holderrith et al. Equivariant Learning of Stochastic Fields: Gaussian Processes and Steerable Conditional Neural Processes. ICML (2021).
>
> [5] FNP: Fourier Neural Processes for Arbitrary-Resolution Data Assimilation. NeurIPS (2024).
>
> [6] Andersson et al. Environmental sensor placement with convolutional Gaussian neural processes. Environmental Data Science 2 (2023).
>
> [7] Bruinsma et al. Autoregressive Conditional Neural Processes. ICLR (2023).
>
> [8] Vaughan et al. Convolutional conditional neural processes for local climate downscaling. Geoscientific Model Development (2022).
>
> [9] Rußwurm et al. Geographic Location Encoding with Spherical Harmonics and Sinusoidal Representation Networks. ICLR (2024).

---

> > ### Author Response · Authors · 2024-11-18
> > **Response to Reviewer XaBC (part 2)**
> >
> > **Response to individual concerns**
> >
> > ***“Honestly, I do not see the value of Figure 1, in particular, since Figure 2 outlines the same pipeline both more detailed and concrete.”***
> >
> > While Figure 2 indeed provides a detailed illustration of our specific model, Figure 1 serves a distinct and important purpose: it presents a novel, unifying framework for understanding conditional neural process (CNP) models—a perspective not previously articulated in the literature.
> >
> > Specifically, Figure 1:
> > 1. Establishes a general construction for CNP architectures based on their encoder, processor, and decoder components.
> > 2. Contextualises our gridded TNP as one instance within this broader family of models.
> > 3. Enables readers to understand how existing CNP variants relate to each other and to our contribution.
> >
> > This high-level framework complements the detailed architecture shown in Figure 2, providing context for understanding where our work fits within the broader landscape of CNPs. Nonetheless, we acknowledge that it has less value than the other figures with respect to our specific approach. We have therefore moved it to the Appendix in the revised version.
> >
> > ***“...but misses a central message and problem that is tackled.”***
> >
> > We respectfully disagree and would like to clarify our paper’s central message, which is consistently presented throughout the manuscript:
> >
> > **The Core Problem:**
> > Current transformer-based models (including TNPs) cannot efficiently process unstructured data using modern attention mechanisms (e.g. ViT and Swin Transformer). This limitation is explicitly stated in:
> > 1. Abstract (lines 15-20).
> > 2. Introduction (lines 65-70).
> > 3. Related Work (Section 3.2, lines 222-224).
> > 4. Methods (Section 4, lines 242-250).
> > 5. Conclusion (Section 6, lines 553-534).
> >
> > **Our Solution:**
> > We present two novel contributions:
> > 1. Transform the unstructured token representations to a structured grid of pseudo-tokens, enabling the use of efficient grid-based attention mechanisms.
> > 2. Develop a method to map these structured representations back to arbitrary unstructured locations.
> >
> > While this methodology could enhance existing applications like the data assimilation step used to create ERA5, our focus is broader: developing a general framework for efficient transformer-based processing of unstructured data. We highlight the example of weather modelling throughout our paper as this serves as a principal motivation for this work. While we do mention this in the paper, we will make it clearer in the revised version.
> >
> > ***“It is hard to assess, whether the introduced method offers advantages over traditional data assimilation methods.”***
> >
> > We clarify that our work’s scope extends beyond data assimilation. Our method addresses a more general problem: mapping from datasets to predictions at arbitrary input locations, where ‘locations’ can represent:
> > 1. Spatial coordinates.
> > 2. Spatio-temporal coordinates.
> > 3. Points in any abstract input space.
> >
> > While this framework can accommodate data assimilation tasks, its generality is a key strength, enabling applications to diverse problems such as:
> > 1. Spatial interpolation e.g. downscaling.
> > 2. Multi-resolution modelling.
> > 3. General regression tasks.
> > 4. Pattern completion.
> > 5. Feature prediction in arbitrary domains.
> >
> > This broader scope is intentional, as it allows our method to serve as a general-purpose tool rather than a specialised solution for data assimilation or forecasting alone. We modified the introduction to more clearly reflect this.
> >
> > ***“I would therefore suggest a focus shift from minimizing forecast error, …, to generating a coherent state of the atmosphere…”***
> >
> > We clarify that minimising forecast error is *not* the focus of our work. Our paper’s primary contribution is methodological: we advance transformer-based models (specifically TNPs) to handle unstructured spatio-temporal data efficiently.
> >
> > Our objectives are:
> > 1. Enable efficient attention mechanisms for unstructured data.
> > 2. Extend the capabilities of TNPs.
> > 3. Create a general-purpose framework applicable across multiple domains.
> >
> > While our method could be applied to tasks like forecasting or atmospheric state estimation, these are potential applications rather than our primary focus. Our goal is to provide fundamental methodological improvements to transformer-based architecture that can benefit a wide range of applications.
> >
> > ***“Manuscript can be improved in terms of supporting claims”***
> >
> > We value this suggestion, and agree that additional support for our claims would strengthen the paper. The broad applicability of NPs has been extensively demonstrated in the literature. For example, see [5], [6] and [8].
> >
> > We will expand our discussion to include these examples, highlighting how they demonstrate NPs’ versatility in modelling dataset-to-prediction mappings across diverse tasks.

---

> > > ### Author Response · Authors · 2024-11-18
> > > **Response to Reviewer XaBC (part 3)**
> > >
> > > ***“The use of different encoders… sounds inspired by ClimaX…”***
> > >
> > > While both our work and ClimaX handle multiple modalities, the approaches are *fundamentally different*:
> > >
> > > **ClimaX:**
> > > 1. Aggregates variable-specific grids through cross-attention with a latent token.
> > > 2. Requires input data to be structured (gridded).
> > >
> > > **Our Approach:**
> > > 1. Maps unstructured observations of each variable to variable-specific grids of pseudo-tokens.
> > > 2. Combines modalities by concatenating pseudo-token representations at each grid location, and passing this pointwise through an MLP.
> > > 3. Explicitly handles unstructured data by design.
> > >
> > > The fundamental difference in handling unstructured data is a key innovation of our work, as emphasised throughout the paper. The ability to process unstructured observations differentiates our method from not only ClimaX, but from other existing approaches.
> > >
> > > ***“...it would be valuable to understand how the introduced method performs on a small dataset, where ANP and TNP are applicable…”***
> > >
> > > This is a valuable suggestion, thank you. We have run additional experiments on a smaller GP regression experiment, which samples the context set size as $N_c \sim U[1, 1000]$. We compare the performance of the Swin-TNP to the TNP. The results, included in the appendix of the revised manuscript (Section E.1.1), demonstrate that the performance difference between the Swin-TNP and full TNP is negligible.
> > >
> > > ***“GraphCast and GenCast follow a similar processing pipeline…”***
> > >
> > > We respectfully disagree: in their current form, GraphCast and GenCast follow fundamentally different processing pipelines from our approach:
> > >
> > > **GraphCast/GenCast:**
> > > 1. Require structured (gridded) input data, specifically ERA5 analysis.
> > > 2. Use fixed connections between observed data and a latent mesh.
> > > 3. Cannot process unstructured observations.
> > > 4. Cannot generate predictions at arbitrary locations.
> > >
> > > **Our Approach:**
> > > 1. Is explicitly designed for unstructured input data.
> > > 2. Dynamically maps unstructured observations to a learned latent grid.
> > > 3. Generates predictions at any location.
> > > 4. Removes the architectural dependency of fixed input grids.
> > >
> > > The ability to handle both unstructured inputs and evaluate at arbitrary locations is a key differentiator of our work, as detailed throughout the paper. While these models are powerful for gridded data, they cannot address the broader class of problems our method tackles.
> > >
> > > ***“A colourmap as well as a difference plot to the ground truth would help…”***
> > >
> > > We have included colourmaps of the predictions and colourmaps of the difference between the predictions and ground-truth in the revised version (see Figures 15 and 16). We are happy to have them in the final version if you think they illustrate the differences between the performance of the models more clearly.
> > >
> > > ***“Similarly, it is difficult to extract the content of Figure 16, which is too dense. Zooming into certain regions could help”***
> > >
> > > We agree that Figure 16 (currently Figure 18) is dense and this is why we chose to show a zoomed-in version in the main text (see Figure 4, zoomed in on the US), and only include Figure 16 in the Appendix.
> > >
> > > ***“Readers might benefit if the prediction task… was introduced as diagnosis”***
> > >
> > > Thank you for this suggestion. While we understand that `diagnosis’ is a common term in meteorological modelling, we believe that maintaining the term ‘prediction’ is more appropriate for our paper for several reasons:
> > > 1. Our tasks involve inferring unknown observations at arbitrary spatio-temporal locations from a set of known observations.
> > > 2. This setup aligns with the standard definition of prediction tasks in the machine learning literature.
> > > 3. Our primary audience is the machine learning community, where this terminology is well-established.
> > >
> > > While we appreciate the meteorological perspective, reframing these tasks as ‘diagnosis’ could potentially obscure their fundamental nature and our paper’s broader machine learning contribution. Our goal is to maintain clarity and precision in communicating with the ICLR audience.

---

> > > > ### Author Response · Authors · 2024-11-18
> > > > **Response to Reviewer XaBC (part 4)**
> > > >
> > > > ***“How is the latent grid structured? … what resolution is chosen and how are the grid points distributed over the sphere…?”***
> > > >
> > > > Our implementation uses a rectilinear grid for pseudo-token placement. Concretely, we apply the following specification:
> > > > 1. Latitude range: $[\lambda_{\min}, \lambda_{\max}]$
> > > > 2. Longitude range: $[\phi_{\min}, \phi_{\max}]$
> > > > 3. Grid size: $(N_1, N_2)$.
> > > > 4. Pseudo-token locations: $\{(\lambda_{\min} + ((\lambda_{\max} - \lambda_{\min}) \times n_1) / N_1, \phi_{\min} + ((\phi_{\max} - \phi_{\min})\times n_2) / N_2)\}$ where $n_1\in 0, \ldots, N_1 - 1$ and $n_2\in 0, \ldots, N_2 - 1$.
> > > >
> > > > We do take into account spherical considerations in our input embeddings for obtaining the initial context and target tokens. Specifically:
> > > > 1. We use spherical harmonic embeddings for each input location $x$ (see [9]).
> > > > 2. We concatenate these embeddings with the observation $y$ and pass through an MLP to obtain initial tokens.
> > > >
> > > > This approach captures spherical geometry. The rectilinear grid locations are used for nearest neighbour calculations and window construction in the Swin Transformer, where we also take into account longitudinal wrapping from 180 to -180.
> > > >
> > > > Finally, we would like to point out that for all experiments and for all models, we provide the grid sizes in our paper (see the grid size column in all tables).
> > > >
> > > > ***“What effect does the size of the attention [window] in the decoder have on the performance?”***
> > > >
> > > > This is an interesting question, and we ran additional experiments to investigate this further in the weather station experiment. The additional experiments are shown in Appendix E.2.1 - Analysis of influence of nearest-neighbour encoding and decoding (highlighted in red), alongside a short discussion about the results.

---

> ### Author Response · Authors · 2024-11-21
> **Reminder of our Rebuttals**
>
> Dear Reviewer XaBC,
>
> We appreciate that you are likely to be reviewing multiple other papers; however, as we are approaching the end of the discussion period (less than one week remaining), we would greatly appreciate your feedback on our rebuttals. Your additional insights would be valuable in helping us improve the paper further.
>
> Kind regards,
> The authors of paper 9931

---

### Author Response · Authors · 2024-11-18
**General Response**

We sincerely thank all reviewers for their thoughtful feedback and valuable suggestions, which have been instrumental in improving the quality of our manuscript. We are delighted that the reviewers recognized the significant "contribution of our paper to the spatiotemporal modelling community" and appreciated our method as a "flexible and elegant way" to handle unstructured data (or the mix of unstructured and structured data).

However, we acknowledge that there may have been some confusion among certain reviewers regarding the primary aim of the paper. Specifically, our goal is to enable TNPs to efficiently process large, unstructured spatio-temporal data by leveraging established efficient attention mechanisms. We have elaborated on this point in our individual responses and addressed all additional concerns and questions raised by each reviewer.

Furthermore, we have uploaded **a revised version of the manuscript** with relevant changes highlighted in red, based on the feedback received. We hope these updates resolve any misunderstandings and manage to convey the main message of the paper more clearly.

---

### Meta-Review · Area_Chair_rsY9 · 2024-12-21

**Metareview:**

The paper presents transformer neural processes with specialized encoders and decoders to handle unstructured inputs and a attention-based processor to take advantage of the transformer scaling performance. They show experiments using ERA5 weather data by randomly sampling the data at sparse locations and using their model to predict temperature and wind at the same locations.

Strengths: A new method to deal with unstructured inputs is valuable to the community, well-organized, clear and detailed manuscript
Weaknesses: The biggest weakness is the limited evaluation of their model on tasks that require unstructured grids - the authors motivate the framework as general, broad, and flexible to a wide range of tasks in SciML or ML in general, but narrow the focus of their results on ERA5 and very simplified weather forecasting. Another smaller weakness is limited discussion of their model in comparison with many other ways to deal with unstructured data

Points for improvement:

Evaluation points:
The evaluation on subsampled ERA5 is positive and appreciated but the paper would have benefitted from stronger evaluations to motivate the contribution. The authors mention: Spatial interpolation e.g. downscaling,  Multi-resolution modelling, General regression tasks, Pattern completion, Feature prediction in arbitrary domains. Many of these have diverse datasets (or easily constructable), especially in PDE modeling (similar to weather forecasting).

Discussion points:
- discussions of GNNs (the request to compare to a GNN based model that includes GraphCast or graph neural operators is still valid since they should be able to process unstructured points, in principle - a discussion is necessary)
- discussions on ViT based models (transformers are designed to process arbitrary sequence of points - it is unclear from the paper why a ViT that processes a sequence of points cannot be used directly with a coordinate based position embedding or similar and needs to be atleast discussed). As noted in ClimaX referenced by one of the reviewers - "Moreover, since the standard ViT treats image modeling as pure sequence-to-sequence problems, it can perform tasks that some other variations cannot, such as learning from spatially incomplete data, where the input does not necessarily form a complete grid. This is useful in the regional forecasting task we consider in Section 4.2.2" - Pg 10 of https://arxiv.org/pdf/2301.10343.

The review points that requested the paper to be designed around data assimilation (replacement of ERA5, which is out-of-scope) or comparison with GenCast (retraining of the GenCast framework with using GraphCast to process an arbitrary mesh) were not used for the final decision.

**Additional Comments On Reviewer Discussion:**

A major concern raised by two reviewers was comparison of the proposed model to SOTA models in weather forecasting. However, the author rebuttals clarify that their contribution is not in competing with SOTA DL weather models but in introducing a methodology to process unstructured grid - they use the weather data as a compelling application to show this.

Another major concern was significance of the work. Reviewer XaBC raised this issue and suggests re-framing of the paper as a data assimilation framework. While this is a steep request and not necessary in itself, the concern is still valid since the experiments are very brief in evaluating a very small set of weather variable forecasting on a random collection of points. The experiment on multimodal inputs is also incremental since it is common for DL frameworks to forecasts hundreds of channels for the ERA5 dataset.

Most reviewers found the positives of the paper to outweigh the negatives (above) and I believe reviewer 7UA7's (score 5) concerns were addressed satisfactorily in the author rebuttal. The only reviewer with a low score (XaBC) did not engage in discussions but was a confident score 3.

I am currently weighing the weaknesses mentioned above in relation to limited evaluations high but I am unopposed to this paper being accepted as well owing to the 2 positive reviews (8 and 6), the fact that the paper is indeed written well with postive results,  as well as the satisfactory rebuttals by the author for the score 5 review.

---

### Decision · Program_Chairs · 2025-01-22

Reject